

# A new model mechanism for atmospheric oxidation of isoprene: global effects on oxidants, nitrogen oxides, organic products, and secondary organic aerosol

Kelvin H. Bates[1] and Daniel J. Jacob[2]

[1]Faculty of Arts and Sciences, Harvard University, Cambridge, MA 02138, USA
[2]School of Engineering and Applied Sciences, Harvard University, Cambridge, MA 02138, USA

**Correspondence:** Kelvin H. Bates (kelvin_bates@fas.harvard.edu)

**Abstract.**

Atmospheric oxidation of isoprene, the most abundantly emitted non-methane hydrocarbon, affects the abundances of ozone ($O_3$), the hydroxyl radical (OH), nitrogen oxide radicals ($NO_x$), carbon monoxide (CO), oxygenated and nitrated organic compounds, and secondary organic aerosol (SOA). We analyze these effects in box models and in the global GEOS-Chem chemical

5   transport model using the new Reduced Caltech Isoprene Mechanism (RCIM) condensed from a recently developed explicit isoprene oxidation mechanism. We find many similarities with previous global models of isoprene chemistry along with a number of important differences. Proper accounting of the isomer distribution of peroxy radicals following the addition of OH and $O_2$ to isoprene influences the subsequent distribution of products, decreasing in particular the yield of methacrolein, and increasing the capacity of intramolecular hydrogen shifts to promptly regenerate OH. Hydrogen shift reactions throughout the

10   mechanism lead to increased OH recycling, resulting in less depletion of OH under low-NO conditions than in previous mechanisms. Higher organonitrate yields and faster tertiary nitrate hydrolysis lead to more efficient $NO_x$ removal by isoprene and conversion to inorganic nitrate. Only 20% of isoprene-derived organonitrates (excluding peroxyacyl nitrates) are chemically recycled to $NO_x$. The global yield of formaldehyde from isoprene is 22% per carbon and less sensitive to NO than in previous mechanisms. The global molar yield of glyoxal is 2%, much lower than in previous mechanisms because of deposition and

15   aerosol uptake of glyoxal precursors. Global production of isoprene SOA is about one third each from isoprene epoxydiols (IEPOX), organonitrates, and tetrafunctional compounds. We find a SOA yield from isoprene of 13% per carbon, much higher than commonly assumed in models, and likely offset by SOA chemical loss. We use the results of our simulations to further condense RCIM into a Mini-Caltech Isoprene Mechanism (Mini-CIM) for less expensive implementation in atmospheric models, with a total size (108 species, 345 reactions) comparable to currently used mechanisms.

## 20  1  Introduction

Isoprene (2-methyl-1,3-butadiene), the dominant hydrocarbon emitted to the atmosphere by plants, plays a central role in tropospheric chemistry. Its global emission is estimated to be $\sim$500 Tg a$^{-1}$, comparable to that of methane (Guenther et al., 2012). Its atmospheric lifetime is only $\sim$1 h against oxidation by the hydroxyl radical (OH), the main tropospheric oxidant.



The high reactivity of isoprene and the subsequent cascade of oxidation products have important implications for tropospheric ozone (Squire et al., 2015), the hydroxyl radical (Lelieveld et al., 2008), the nitrogen cycle (Paulot et al., 2013), and secondary organic aerosol (SOA) (Carlton et al., 2009). The persistence of long-lived oxidation products extends isoprene's influence to regional and global scales (Kanakidou et al., 2005; Paulot et al., 2012).

Proper representation of isoprene chemistry is of critical importance for global models of atmospheric chemistry, but the mechanism is complicated and models often use outdated information. Wennberg et al. (2018) presented a detailed review of current knowledge and compiled a comprehensive mechanism. This mechanism is far too complex for implementation in atmospheric models, but Wennberg et al. (2018) also compiled a reduced version suitable for the range of conditions found in the atmosphere. We examine here its implications for the range of effects of isoprene on atmospheric chemistry.

The isoprene oxidation cascade varies considerably depending on local atmospheric conditions. Different branches in the chemical mechanism develop depending on the reactions of the peroxy radicals ($RO_2$) produced in the initial and subsequent oxidation steps. Reaction with NO produces ozone and organic nitrates, and reactions of acylperoxy radicals with $NO_2$ produce peroxyacyl nitrates (PANs). Reactions with $HO_2$ are typically OH-consuming via hydroperoxide formation. Intramolecular hydrogen shift (H-shift) reactions tend to propagate radical chains and regenerate OH. These different branches of $RO_2$

chemistry also produce a large and differing ensemble of oxygenated multifunctional compounds, some of which have low volatility and/or aqueous-phase chemistry leading to SOA formation.

    Isoprene oxidation mechanisms in atmospheric models have evolved considerably over the past decades. Early mechanisms focused on high-NO conditions representative of polluted regions and the role of isoprene in driving the production of ozone and organic nitrates (Lloyd et al., 1983; Brewer et al., 1984; Trainer et al., 1987; Madronich and Calvert, 1990). These model

studies led to a number of chamber experiments to test and improve the mechanisms (Atkinson et al., 1989; Tuazon and Atkinson, 1990; Paulson et al., 1992; Paulson and Seinfeld, 1992; Grosjean et al., 1993; Miyoshi et al., 1994; Kwok and Atkinson, 1995; Stevens et al., 1999; Ruppert and Heinz Becker, 2000) along with field observations of isoprene chemistry (Biesenthal and Shepson, 1997; Starn et al., 1998; Roberts et al., 1998; Wiedinmyer et al., 2001). Improved understanding of the chemistry under low-NO conditions and growing interest in formation of organic aerosol led to the development of

increasingly complex mechanisms (Carter, 1996; Stockwell et al., 1997; Pöschl et al., 2000; Geiger et al., 2003; Aumont et al., 2005). The regularly updated Master Chemical Mechanism (MCM) presents a nearly explicit compilation of isoprene chemistry (Jenkin et al., 1997; Saunders et al., 2003; Jenkin et al., 2015). Various versions of these mechanisms have been incorporated into atmospheric models (Fan and Zhang, 2004; Pfister et al., 2008; Taraborrelli et al., 2009; Archibald et al., 2010; Mao et al., 2013; Squire et al., 2015; Chan Miller et al., 2017; Müller et al., 2018), with no coalescence toward a unified

mechanism across models.

    The effect of isoprene on OH concentrations has elicited much controversy. Early mechanisms exhibited near-complete titration of OH by isoprene under low-NO conditions (Jacob and Wofsy, 1988). However, this was contradicted in the early 2000s by observations of elevated OH concentrations in tropical forests (Carslaw et al., 2001; Lelieveld et al., 2008; Martinez et al., 2010; Pugh et al., 2010; Whalley et al., 2011; Stone et al., 2011). Models attempted to correct for this behavior by

invoking hypothetical OH-recycling mechanisms in their low-NO oxidation schemes (Butler et al., 2008; Taraborrelli et al.,





2009; Archibald et al., 2010; Kubistin et al., 2010). These were later replaced with mechanistic OH-recycling pathways, including isoprene epoxydiol (IEPOX) formation (Paulot et al., 2009b) and H-shift chemistry (Peeters et al., 2009; Asatryan et al., 2010; Crounse et al., 2011).

Another focus of interest has been the role of isoprene as a sink for $NO_x$ ($\equiv NO + NO_2$) through organonitrate formation
and the subsequent fates of these organonitrates (von Kuhlmann and Lawrence, 2004; Horowitz et al., 2007; Wu et al., 2007; Paulot et al., 2012). Differences between models in organonitrate yields and recycling have large effects on simulated ozone (Fiore et al., 2012; Xie et al., 2013; Müller et al., 2014). Recent work has established that hydrolysis and deposition of isoprene-derived organonitrates can be a dominant $NO_x$ loss process in some environments (Romer et al., 2016; Fisher et al., 2016).

The role of isoprene as a SOA precursor has received increasing interest following evidence from field studies of $C_5$ com-
pounds in ambient particles (Claeys et al., 2004; Edney et al., 2005; Kleindienst et al., 2007), which led to experimental work measuring SOA yields from isoprene oxidation (Kroll et al., 2005, 2006; Surratt et al., 2006). Further investigations of the isoprene oxidation mechanism identified specific SOA precursors such as IEPOX (Paulot et al., 2009b; Nguyen et al., 2014), oxidation products of methacryloyl peroxynitrate (MPAN) (Nguyen et al., 2015a), and highly functionalized compounds with low volatility (Krechmer et al., 2015; D'Ambro et al., 2017). These SOA formation pathways are now commonly implemented
in models (Marais et al., 2016; Stadtler et al., 2018).

Here we implement the condensed version of the Wennberg et al. (2018) comprehensive mechanism, which we call the Reduced Caltech isoprene mechanism (RCIM), in three types of atmospheric models – a fixed-radical box model, a diurnal-steady-state box model, and the global GEOS-Chem chemical transport model. RCIM introduces a number of components not currently included in the models, as described in Section 2. We use it to investigate the effects of isoprene chemistry on
hydrogen oxide radicals ($HO_x \equiv OH$ + peroxy radicals), $NO_x$, and ozone. We also investigate the fate of the isoprene carbon, including the yields of oxygenated organic products, CO, $CO_2$, and SOA. In the process we compare RCIM to previous isoprene oxidation mechanisms, including MCM v3.3.1, the current GEOS-Chem standard mechanism (v11-02c), and others in the literature. Finally, we use the results of our simulations to further simplify the isoprene mechanism for computational savings in model applications.

## 2   Methods

### 2.1   Chemical mechanism

RCIM is v4.1 of the "Reduced-plus" mechanism found in the Wennberg et al. (2018) mechanism repository (DOI 10.7907/ Z9S75DHB). It includes the oxidation of isoprene by OH, ozone, and $NO_3$ and condenses the ensuing oxidation cascade for the practical range of atmospheric conditions. The mechanism includes 148 species and 412 reactions representing the
complete isoprene oxidation cascade (full conversion to $CO_2$), in contrast to the 385 species and 810 reactions in the full explicit mechanism.

The reader is directed to Wennberg et al. (2018) for a detailed description of RCIM. Most importantly, the mechanism treats the initial system of allylic and peroxy radicals formed following the addition of OH to isoprene dynamically, as shown in





Figure 1. Older mechanisms implicitly used fixed distributions of isoprene-hydroxy-peroxy (ISOPOO) radicals, often derived from experiments performed under high-NO conditions. Addition of $O_2$ to allylic radicals under ambient conditions is in fact a reversible process, resulting in a dynamic system with differing initial (kinetic) and equilibrium radical distributions (Teng et al., 2017). In RCIM, we simplify this ten-species, 69-reaction radical system to two species as shown in the bottom panel

of Figure 1 and described in Wennberg et al. (2018). The implications of this novel treatment of the isoprene-hydroxy-peroxy radical system are manifest throughout this paper but are discussed specifically in Section 3.3.

Additional aspects of RCIM relative to older mechanisms include: new products and decreased $C_5$-hydroperoxy-aldehyde (HPALD) yields following the 1,6-H-shifts of the $Z$-$\delta$-OH-peroxy radicals shown in Figure 1 (Teng et al., 2017); more intramolecular H-shifts, including rapid peroxy-hydroperoxy shifts (Jørgensen et al., 2016; Möller et al., 2019), resulting

in higher OH recycling under low-NO conditions; new parameterizations of nitrate yields from $RO_2$ + NO reactions, including pressure and temperature dependence; explicit treatment of highly functionalized products such as $C_5$-dihydroxy-dihydroperoxides and other tetrafunctionalized $C_5$ compounds; and more detailed chemistry following the reactions of isoprene with $NO_3$ (Schwantes et al., 2015) and ozone (Nguyen et al., 2016).

In this paper we compare RCIM to MCM v3.3.1 (Jenkin et al., 2015) and to the current standard GEOS-Chem mechanism

implemented in version 11-02c. MCM v3.3.1 (subsequently MCM for short) treats the chemistry of isoprene more explicitly than RCIM or the v11-02c mechanism. It includes 602 species and 1926 reactions. The GEOS-Chem v11-02c chemical mechanism includes 106 species and 335 reactions, and is primarily derived from chamber experiments separating the high- and low-NO oxidation schemes of isoprene (Mao et al., 2013; Paulot et al., 2009a, b) with targeted updates to nitrate yields, peroxy radical H-shift chemistry, and IEPOX chemistry (Bates et al., 2014, 2016; Travis et al., 2016; Fisher et al., 2016; Marais et al.,

2016; Chan Miller et al., 2017). The GEOS-Chem mechanism lumps the initial isoprene peroxy radicals into a single ISOPOO species.

Neither RCIM nor MCM include the loss of organic products by deposition or SOA formation. We exclude these processes from box model simulations, so as to isolate the effects of gas phase chemistry. For global GEOS-Chem simulations with RCIM, we extend the existing deposition (Nguyen et al., 2015b; Travis et al., 2016) and SOA formation (Marais et al.,

2016) parameterizations from GEOS-Chem v11-02c to analogous species in the mechanism. All $C_5$ epoxides are thus treated identically to IEPOX, tetrafunctional $C_5$ nitrates are treated identically to isoprene dihydroxy-dinitrates, and tetrafunctional $C_5$ non-nitrates are treated identically to the "LVOC" low-volatility species that represents this group of SOA precursors in GEOS-Chem v11-02c. The one major change we implement relative to the standard v11-02c mechanism is to increase the aerosol uptake rate for tertiary nitrates by a factor of ten, following evidence that the hydrolysis of these nitrates proceeds

rapidly in the atmosphere (Darer et al., 2011; Hu et al., 2011; Rindelaub et al., 2014; Xu et al., 2015; Rindelaub et al., 2016).

## 2.2 Simulations

All models use the same fast Rosenbrock kinetic solver implemented with the Kinetic Preprocessor tool (KPP; Damian et al., 2002; Daescu et al., 2003; Sandu et al., 2003).





**Fixed-radical box modeling:** In order to quantify product yields per unit of isoprene oxidized by OH under a given set of atmospheric conditions, a series of box model simulations are conducted with fixed concentrations of NO and $HO_2$. This method serves to remove most nonlinearities and feedbacks inherent in the isoprene oxidation mechanism, so as to isolate the effects of the radicals on the oxidation pathways. The model is initialized with 1 ppbv of isoprene and run until complete

conversion to $CO_2$. (In the absence of deposition or SOA formation, all isoprene carbon is eventually converted to $CO_2$). The branching pathways of isoprene oxidation products are computed assuming 0.1 pptv OH, an $NO_2$/NO ratio of 5, no ozone or $NO_3$, temperature of 298.15 K, and solar radiation for clear-sky equatorial midday with an ozone column of 350 DU. Additional sensitivity simulations with differing temperature and photolysis settings, along with simulations investigating isoprene + $NO_3$ chemistry, can be found in Sections S2 and S4 of the Supplement, respectively.

**Diurnal-steady-state box modeling:** Additional box model simulations are run with variable radical concentrations and diurnal cycles of temperature, sunlight, and isoprene emissions. These simulations follow the setup and conditions of Jenkin et al. (2015) for MCM to facilitate comparison. The box model simulates a tropical continental boundary layer with isoprene and NO emissions, ventilated by the background free troposphere with a fixed exchange rate constant corresponding to a ventilation time scale of 1 day. The free tropospheric background includes 1.8 ppmv $CH_4$, 100 ppbv CO, 20 ppbv ozone,

300 pptv formaldehyde, and 1% $H_2O$. Isoprene emissions vary diurnally with both temperature and sunlight, as parameterized by Guenther et al. (1995), for an average daytime emission rate of $7.6 \times 10^6$ molecules $cm^{-3}$ $s^{-1}$ (Eerdekens et al., 2009). Photolysis rates follow the diurnal pattern for clear sky at the Equator with an ozone column of 350 DU. Temperature follows a sinusoidal diurnal pattern with an amplitude of 4 °C, peaking at 13:00. Simulations are initialized for seven days, after which concentrations from the eighth day (daytime averages between 06:00 and 18:00) are used in the results reported below.

**Global modeling:** We incorporate RCIM into the GEOS-Chem global 3-D model (http://geos-chem.org). GEOS-Chem is driven by assimilated meteorological observations from the NASA Goddard Earth Observing System – Fast Processing (GEOS-FP) of the NASA Global Modeling and Assimilation Office (GMAO). We use UCX version 11-02c as a base, including both tropospheric and stratospheric chemistry (Eastham et al., 2014) and with tropospheric methane fixed on the basis of observations. Emissions use the standard HEMCO configuration in v11-02c (Keller et al., 2014), including isoprene emissions

from the MEGAN v2.1 inventory (Guenther et al., 2012) as implemented into GEOS-Chem by Hu et al. (2015) and scaled uniformly to 535 Tg $a^{-1}$ (Guenther et al., 2012). Annual isoprene emissions are shown in Figure 2. We conduct simulations for 1 year (July 2014 – June 2015) following 1.5 years of initialization starting in January 2013. Baseline simulations are conducted at $2° \times 2.5°$ horizontal resolution with 72 vertical levels and additional sensitivity simulations are conducted at $4° \times 5°$ horizontal resolution. We find no significant differences in results between the two resolutions, consistent with a previous GEOS-Chem

investigation of the effects of model resolution on isoprene chemistry (Yu et al., 2016). For regional-scale results, we use the outputs of $2° \times 2.5°$ horizontal resolution simulations at 0 - 1 km altitude and average over 81.25 - 93.75 °W, 31 - 39 °N for the Southeast United States, 53.75 - 76.25 °W, 11 °S - 3 °N for the Amazon Basin, and 111.25 - 121.25 °E, 23 - 41 °N for East China.





## 3 Isoprene oxidation pathways

### 3.1 Initial oxidant branching

Figure 3 (left panels) shows the contributions of OH, ozone, and $NO_3$ to the overall oxidation of isoprene using RCIM in diurnal-steady-state box model simulations and in GEOS-Chem. On a global scale, we find that 88% of isoprene is oxidized

by OH, 10% by ozone, and 1.7% by $NO_3$. These global averages mask some spatial variability, as shown in Figure 4 and Table 1; for example, $NO_3$ oxidation contributes up to 5% of isoprene loss locally in the Southeast United States, and oxidation by ozone contributes up to 15% over tropical forests.

Although the ozone and $NO_3$ oxidation pathways represent relatively minor contributions to global isoprene oxidation, they can be important for the global budgets of certain compounds. For example, a sensitivity GEOS-Chem simulation without

isoprene ozonolysis results in a 51% global mean decrease in formic acid production from isoprene and a 25% decrease in hydroxymethyl-methyl-$\alpha$-lactone, a product of methacrolein oxidation and SOA precursor (Nguyen et al., 2015a). Similarly, a GEOS-Chem simulation without isoprene + $NO_3$ results in a 39% decrease in isoprene-derived $C_{2+}$ organonitrate concentrations. Additional results from simulations excluding the ozone and $NO_3$ pathways can be found in Table S1.

RCIM results in a higher contribution of isoprene + OH to the total oxidation of isoprene than past estimates, as shown in

Table 1, and lower contributions from ozone and $NO_3$. We ascribe this change primarily to increased OH recycling relative to older mechanisms (see Section 4). This further explains the temperature dependence of the OH pathway contribution in the diurnal-steady-state box model (Figure 3, top left), which is also stronger than in past mechanisms, as a result of the temperature dependence of OH-recycling H-shift reactions. Nighttime oxidation by $NO_3$ is particularly lower than previously reported in the literature (5-7%, Table 1), which largely reflects the amount of isoprene remaining at sunset. More efficient

recycling of OH in RCIM would result in less isoprene at sunset.

### 3.2 Fate of ISOPOO

The ISOPOO radicals produced following the reaction of isoprene with OH and addition of $O_2$ (Figure 1) represent an important branching point in the isoprene oxidation cascade. The relative contributions of ISOPOO's reactions with NO, $HO_2$, $RO_2$, and via unimolecular H-shifts largely set the chemical outcomes of the oxidation mechanism, including ozone formation, OH

titration vs. recycling, and SOA production.

The top central panel of Figure 3 shows the relative contributions of each pathway as a function of $NO_x$ and temperature in diurnal-steady-state box model simulations. At a mean temperature of 25 °C, reaction with NO dominates the ISOPOO fate at $NO_x$ > 500 pptv and reaction with $HO_2$ dominates at lower $NO_x$, while H-shifts account for up to 30% at low NO. H-shift rates are strongly temperature-dependent (Teng et al., 2017), and reach a comparable importance to reaction with $HO_2$ at 35

°C.

The global contributions of each pathway as computed by GEOS-Chem are summarized in Table 1. While reaction with $HO_2$ represents the dominant fate of ISOPOO in the atmosphere, reaction with NO and H-shifts contribute major portions of the global total. We estimate a 50% larger flux through the H-shift pathway than in the GEOS-Chem v11-02c mechanism; this



contributes to the higher OH recycling in RCIM (Section 4.1). Our estimated fraction of ISOPOO undergoing H-shifts is 25% lower than that of Peeters et al. (2014), which used faster 1,6-H-shift rates, and similar to the recent estimate of Müller et al. (2018).

Figure 4 illustrates the global distribution of the ISOPOO fate. The NO pathway dominates in polluted regions of northern midlatitudes, contributing up to 50% of ISOPOO reactivity in the Southeast United States and 90% in East China. In remote tropical forests, where low $NO_x$ and $HO_x$ lead to peroxy radical lifetimes in excess of 100 s, the H-shift pathway can account for up to 45% of the ISOPOO fate, and reaction with $RO_2$ can contribute up to 20%.

## 3.3 ISOPOO isomer distributions

The right panels of Figure 3 show the fractional contributions of the ISOPOO isomers to total ISOPOO reactivity in the diurnal-steady-state and global simulations. The $\delta$-ISOPOO isomers are largely responsible for $HO_x$ recycling via rapid unimolecular H-shifts, while the two $\beta$ isomers lead to different subsequent product formation. Methyl vinyl ketone (MVK) is the major product of $\beta$-1-OH-ISOPOO + NO or $RO_2$, while methacrolein (MACR) is the main product from the equivalent $\beta$-4-OH-ISOPOO reactions. MACR leads to SOA formation via the production of hydroxymethyl-methyl-$\alpha$-lactone (HMML) (Nguyen et al., 2015a), while MVK does not.

The $\delta$ isomers comprise a higher fraction of total ISOPOO reactivity under low-NO conditions (up to 30% at 10 pptv $NO_x$ and a mean diurnal temperature of 25 °C) due to the importance of the rapid 1,6 H-shifts of the $Z$-$\delta$ isomers. Within the ISOPOO pool derived from OH addition to isoprene at C4 (comprising 37% of the total; see numbering in Figure 1), the $\delta$ isomer contribution is even higher (up to 57%), due to the more rapid 1,6 H-shift of $Z$-$\delta$-4-OH-ISOPOO. The $\delta$ isomers account for <10% of ISOPOO reactivity at NO > 10 ppbv. At even higher concentrations of NO bimolecular reactivity can be sufficiently high to trap the ISOPOO isomer distribution at its kinetic limit, leading to much higher $\delta$ isomer abundances. Here RCIM deviates substantially from the explicit model, but these conditions are not of general atmospheric relevance.

Table 1 and the bottom-right panel of Figure 3 show the isomers' contributions to total ISOPOO reactivity from GEOS-Chem simulations. We find that the $\delta$ isomers contribute 21% of the total ISOPOO reactivity on a global annual average. This contribution increases in areas with high ISOPOO H-shift fractions (e.g. to 30% in the Amazon), and decreases in areas with higher NO (e.g. to 17% in the Southeast United States).

In the GEOS-Chem v11-02c and older mechanisms, which treat the ISOPOO system as a single species, H-shifts deplete the whole ISOPOO pool, rather than preferentially depleting the 4-OH-ISOPOO radicals. As a result, the fraction of ISOPOO that go on to react bimolecularly, which should be enriched in 1-OH-ISOPOO, is instead assigned the same initial 1-OH:4-OH ratio it would have without H-shifts. This leads to far higher contributions from the 4-OH pathway – up to 58% over the Amazon in the v11-02c mechanism.

The most prominent effect of this change is in the unique subsequent products of the 1-OH and 4-OH systems. By preferentially depleting the 4-OH-ISOPOO pool, H-shifts predominantly remove the potential for formation of MACR and the secondary $\beta$-hydroxynitrate, while the much slower 1,6 H-shift of 1-OH $Z$-$\delta$-ISOPOO has a smaller effect on potential MVK and tertiary $\beta$-hydroxynitrate formation from the 1-OH-ISOPOO system. In RCIM, increasing H-shift contributions thus in-



crease the MVK/MACR and tertiary/secondary nitrate ratios, while in GEOS-Chem v11-02c and older mechanisms these ratios are unaffected by H-shift chemistry. Global simulations with the single-radical ISOPOO representation of Mao et al. (2013) and with the fixed ISOPOO distribution of Paulot et al. (2009a) (while leaving the rest of the chemistry unchanged from RCIM) result in 21% and 18% decreases in tropospheric production of MVK relative to RCIM, respectively, and 24% and 10%

increases in tropospheric production of MACR, approximately doubling the MVK/MACR ratio under low-NO conditions. The dynamic ISOPOO system also results in 25% higher tropospheric production of the tertiary $\beta$-hydroxynitrate in RCIM relative to the single-radical and fixed-distribution ISOPOO representations of Mao et al. (2013) and Paulot et al. (2009a). The rapid hydrolysis of this tertiary nitrate in turn leads to more efficient $NO_x$ removal by isoprene nitrates (see Section 5.2). Table S1 shows additional results of GEOS-Chem simulations with fixed-distribution and single-radical ISOPOO chemistry.

## 10   4   Effects on radical families and ozone

### 4.1   Effects on $HO_x$ radicals

Figure 5 shows the effects of isoprene on OH and $HO_2$ concentrations in diurnal-steady-state box model simulations with the RCIM, MCM, and v11-02c mechanisms. OH is depleted under low-NO conditions by direct reactions with isoprene and its oxidation products. The effect reverses under high-$NO_x$ conditions when these reactions compete with the reaction of $NO_2$ with

OH that is the dominant $HO_x$ sink. Isoprene chemistry enhances $HO_2$ concentrations under all conditions because of photolysis of formaldehyde and other carbonyls producing $HO_x$ radicals, and particularly under high-$NO_x$ conditions by competing with the $NO_2$ + OH reaction. OH depletion from isoprene oxidation under low-NO conditions is strongly temperature-dependent in RCIM and less pronounced than in previous mechanisms. In diurnal-steady-state simulations at $< 100$ pptv $NO_x$, we find that an increase in mean diurnal temperature of $10\,^{\circ}C$ causes up to a doubling in daytime OH concentrations, and that RCIM

sustains OH concentrations twice as high as MCM and three times higher than GEOS-Chem v11-02c.

Differences with MCM and GEOS-Chem v11-02c are due to updated H-shift chemistry in RCIM, which efficiently recycles $HO_x$ as shown in Figure 6. The initial H-shift of the Z-$\delta$-4-OH-ISOPOO radical (the dominant ISOPOO H-shift pathway) is highly temperature-dependent and produces 2.2 equivalents of $HO_x$ (1.5 OH + 0.7 $HO_2$) concurrently with the first generation of stable products. Assuming that photolysis is the dominant fate of the conjugated hydroperoxy-aldehydes (HPALDs) that

make up 60% of the stable products, $HO_x$ production can increase to 3.0 equivalents (2.2 OH + 0.75 $HO_2$ + 0.04 $RO_2$) in the second oxidative generation. Such a regeneration of $HO_x$ is necessary to reconcile models and measurements in low-NO conditions (Fuchs et al., 2013; Kaser et al., 2015; Feiner et al., 2016; Kaiser et al., 2016; Mao et al., 2018).

Figure 7 and Table 2 show the effects of isoprene oxidation as simulated in GEOS-Chem. The global annual mean tropospheric concentration of OH decreases by 11% and that of $HO_2$ increases by 6.5%. OH decreases are largest in tropical

continental boundary layers, but diffuse effects extend throughout the global troposphere due to the influence of longer-lived isoprene oxidation products, in particular CO. Thus the globally integrated effects in the upper troposphere (5-10 km) are comparable to those in the boundary layer (0-1 km). We calculate that isoprene chemistry causes a 12% increase in the tropospheric



lifetime of methane with respect to oxidation by OH, thus enhancing the climatological effects of an already potent greenhouse gas.

As in the diurnal-steady-state simulations, the titration of OH by isoprene oxidation under low-NO conditions is substantially weaker in global simulations with RCIM than with the GEOS-Chem v11-02c mechanism. Whereas isoprene oxidation in the

v11-02c mechanism causes reductions in annual mean OH of ∼90% over the Amazon and Congo basins, RCIM exhibits only ∼70% reductions, sustaining annual mean OH concentrations up to three times higher. Again, this change is largely due to increased $HO_x$ production from the H-shifts of ISOPOO in RCIM. For additional comparison to the MCM and GEOS-Chem mechanisms, see Section S5.2 of the Supplement.

## 4.2   Effects on $NO_x$

Table 2 and Figure 7 summarize the effects of RCIM isoprene chemistry on tropospheric $NO_x$. These effects largely involve the formation and fate of PANs and other organonitrates, and changes in $NO_x$ lifetime due to changes in $HO_x$. On a global annual average, isoprene chemistry depletes tropospheric $NO_x$ by 4.2%. $NO_x$ depletion reaches 50% in tropical continental regions where high VOC/$NO_x$ ratios promote $NO_x$ titration by organonitrate formation. $NO_x$ increases by up to 10% in remote regions such as the oceanic free troposphere due to release of $NO_x$ from transported PANs and other organonitrates.

The effects described above have been shown in past models (von Kuhlmann and Lawrence, 2004; Ito et al., 2009; Fischer et al., 2014; Jenkin et al., 2015), which calculated similar magnitudes for the overall contribution of isoprene to $NO_x$ and nitrate budgets. One significant difference that we find in comparison with the GEOS-Chem v11-02c mechanism is the composition of the organonitrate pool and its effects on $NO_x$ transport and removal. For example, due to higher formation of tertiary nitrates and their increased hydrolysis rate in RCIM, we estimate tropospheric $NO_x$ loss to hydrolysis of nitrates to be 4.9 TgN $a^{-1}$,

compared to only 1.8 TgN $a^{-1}$ with the GEOS-Chem v11-02c mechanism. This increased $NO_x$ loss rate in RCIM is offset by smaller overall organonitrate production and a substantial reduction in the formation of MPAN in low-NO conditions. Thus, the two mechanisms simulate a nearly identical tropospheric $NO_x$ burden, but there are strong distributional differences (see Figure S17). The fate of organonitrates including the fraction recycled as $NO_x$ will be discussed in more detail in Section 5.2.

## 4.3   Effects on ozone

The bottom panel of Figure 5 shows the effect of isoprene oxidation on ozone in diurnal-steady-state box model simulations. Isoprene has little effect under low-NO conditions but stimulates ozone production at higher NO due to increased peroxy radical concentrations, accelerating cycling of NO to $NO_2$. At very high NO, ozone production becomes VOC-limited and the effect of isoprene becomes very large. These dependences are relatively invariant with temperature and similar to those seen in MCM and GEOS-Chem v11-02c.

Figure 7 and Table 2 summarize the effect of isoprene chemistry on ozone as simulated in GEOS-Chem. While isoprene oxidation decreases boundary layer ozone over the Amazon by 22% (3.4 ppbv), mainly because of $NO_x$ depletion, it causes an overall increase in the annual average tropospheric ozone burden of 4.2% (1.9 ppbv), and local increases of up to 6 ppbv in China where ozone production is often VOC-limited (Jin and Holloway, 2015). The release of $NO_x$ from isoprene-derived





organonitrates extends these effects to the free troposphere, with stronger ozone enhancement in the Northern Hemisphere where NO is higher. These results are consistent with past studies diagnosing the influence of isoprene and its oxidation mechanism on ozone (Wang and Shallcross, 2000; von Kuhlmann and Lawrence, 2004; Squire et al., 2015), and investigating the effects of changing isoprene emissions (Sanderson et al., 2003; Wiedinmyer et al., 2006; Ganzeveld et al., 2010; Wu et al.,

2012; Pacifico et al., 2012; Squire et al., 2014).

However, certain new aspects of RCIM cause slight distributional changes in the effects of isoprene on ozone relative to past mechanisms. RCIM's higher first-generation nitrate yields and faster tertiary nitrate hydrolysis relative to the GEOS-Chem v11-02c mechanism lead to reduced ozone formation from the ISOPOO + NO pathway. In the Southeast United States, where past mechanisms have tended to overestimate surface ozone (Murazaki and Hess, 2006; Yu et al., 2007; Lin et al., 2008; Fiore

et al., 2009; Yu et al., 2010; Rasmussen et al., 2012; Travis et al., 2016), RCIM results in a 1.5% decrease in annual mean boundary layer ozone relative to the GEOS-Chem v11-02c mechanism. Reduced $NO_x$ transport in RCIM and reduced MPAN formation under low-NO conditions also results in higher sustained ozone over the Amazon (+3 ppbv) and lower ozone in the remote Southern Hemisphere (-1.5 ppbv) than the GEOS-Chem v11-02c mechanism. For more detailed comparisons with MCM and GEOS-Chem v11-02c, see Section S5.2 of the Supplement.

## 5   Isoprene oxidation products

RCIM is carbon-conserving and can be used to track the fate of isoprene-derived carbon. Figure 8a shows global mean results for the fate of isoprene carbon in GEOS-Chem. 50% of isoprene carbon is oxidized fully to $CO_2$ in the gas phase, in good agreement with the value of 52% calculated by Safieddine et al. (2017) for all non-methane VOCs. 37% is lost to wet and dry deposition of organic oxidation products before full conversion to $CO_2$. This falls between previous estimates of 32% (Müller

et al., 2018) and 44% (Safieddine et al., 2017) for all oxidized non-methane VOCs. The final 13% forms isoprene SOA, which represents a terminal sink in GEOS-Chem (Marais et al., 2016).

The following subsections describe the fate of isoprene-derived carbon and its organic products in greater detail: oxygenated gas-phase VOCs for which observations are available (Section 5.1), organonitrates (Section 5.2), and SOA precursors (Section 5.3). Figure 9 shows the annual average spatial distributions of important isoprene oxidation products in GEOS-Chem. Figure

10 shows molar yields of individual products from OH-initiated oxidation as a function of NO and $HO_2$ in fixed-radical box model simulations. Figure 11 shows daytime mean concentrations of major classes of isoprene products as a function of $NO_x$ and temperature in diurnal-steady-state simulations. Additional details on the organic products of isoprene oxidation are provided in the Supplement, including simulated global and regional molar yields (Table S3), the contributions of specific oxidation pathways to global production (Tables S1 and S2), sensitivities of yields to light and temperature (Section S2), molar

yields from $NO_3$-initiated oxidation (Section S4), and comparisons to MCM and GEOS-Chem v11-02c (Section S5).



## 5.1 Oxygenated VOCs and CO

**CO:** We find the molar yield of carbon monoxide from isoprene oxidation to be 190% globally, or 38% per carbon. Of the 50% of isoprene carbon oxidized fully to $CO_2$, 76% proceeds via CO (Figure 8a). The global CO source from isoprene oxidation is 415 Tg a$^{-1}$, which in GEOS-Chem represents 17% of the total atmospheric CO source (including 36% from methane, 8% from the oxidation of other VOCs, and 39% from direct emissions). Our simulated global molar CO yield from isoprene is slightly higher than the range of 100-170% from previous estimates (Miyoshi et al., 1994; Bergamaschi et al., 2000; Duncan et al., 2007; Pfister et al., 2008), but only a small increase from the 180% molar yield with the GEOS-Chem v11-02c mechanism. Isoprene oxidation generates up to 100 ppbv of CO locally over tropical forests.

**Formaldehyde:** Formaldehyde is measurable by satellites and has been used in this manner to infer isoprene emissions based on estimated formaldehyde yields from isoprene (Palmer et al., 2003; Marais et al., 2012; Kaiser et al., 2018). Marvin et al. (2017) and Wolfe et al. (2016) found from field observations that most current mechanisms underestimate formaldehyde yields from isoprene. Fixed-radical box models with RCIM show 140-190% molar yields (28-38% per-carbon yields) of formaldehyde in the complete gas-phase oxidation of isoprene by OH under atmospherically relevant NO and $HO_2$ concentrations, with increasing yields at higher NO. The variation with NO is less steep than in past mechanisms, exhibiting higher formaldehyde production under low-NO conditions, much of it from the photolysis of $C_4$-dihydroperoxy-aldehydes produced by H-shift chemistry (top-right corner of Figure 6), and lower production under high-NO conditions (see Figures S20 and S24 in the Supplement). While this difference has regional impacts, e.g. increasing the mean annual boundary layer formaldehyde burden by 50% over the Amazon from GEOS-Chem v11-02c, it does not substantially change the overall global molar yield of formaldehyde from isoprene, which we estimate to be 111% (22% per carbon, a 4% increase from GEOS-Chem v11-02c). We find that isoprene contributes 18% of the global formaldehyde burden (Table 2), in line with previous estimates (Pfister et al., 2008).

**Glyoxal:** Glyoxal ($C_2H_2O_2$) is also measured by satellites (Vrekoussis et al., 2009; Alvarado et al., 2014; Chan Miller et al., 2014), and different yields relative to formaldehyde can discriminate between emissions of different VOCs (Chan Miller et al., 2016). Past mechanisms have provided differing estimates on which isoprene oxidation pathways produce the most glyoxal (Li et al., 2016), and comparisons with field measurements show that glyoxal production is higher under low-NO conditions than most mechanisms predict (Li et al., 2016; Chan Miller et al., 2017). RCIM yields of glyoxal from isoprene peak at 10% under low-NO conditions (Figure 10), reflecting contributions from ISOPOO H-shifts and the degradation of IEPOX. In contrast, glyoxal yields in MCM are highest at high NO. Our diurnal steady-state box model simulations show that the RCIM glyoxal/formaldehyde ratio remains in the 2-3% range over the ensemble of atmospheric conditions (see Figure S21), in line with field observations for isoprene-dominated environments (Kaiser et al., 2015; Chan Miller et al., 2017). We find in GEOS-Chem that many glyoxal precursors (IEPOX, nitrates, and tetrafunctional $C_5$ compounds) are lost to aerosol or deposition before they can react in the gas phase, depressing the glyoxal yield relative to the box model simulations where aerosol/deposition effects are not included. This results in a global glyoxal yield from isoprene of 2% in GEOS-Chem with RCIM, only half that reported recently by Müller et al. (2018) and even lower than in some past simulations (Fu et al., 2008;



Myriokefalitakis et al., 2008; Taraborrelli et al., 2009). We find a reduction in global tropospheric glyoxal loading of 60% relative to the GEOS-Chem v11-02c mechanism. However, Chan Miller et al. (2017) found good agreement between glyoxal simulated by GEOS-Chem v11-02c and aircraft observations in the Southeast United States. This suggests that RCIM may underestimate glyoxal yields from isoprene.

**Organic acids:** In GEOS-Chem simulations with RCIM, isoprene contributes 21 Tg a$^{-1}$ of formic acid (a 5.8% molar yield) and 25 Tg a$^{-1}$ of acetic acid (a 5.5% molar yield) globally. Over half of this formic acid comes from the initial reaction of isoprene with ozone, either directly from the stabilized $C_1$ Criegee intermediate (Nguyen et al., 2016) or secondarily through the reaction of hydroxymethyl hydroperoxide with OH (Allen et al., 2018), while the rest is formed in the ozonolysis of MVK and MACR or the reactions of MVK-derived enols and nitrates with OH. Acetic acid is produced in the reactions of $HO_2$ and

$RO_2$ with the acylperoxy radical, a fragmentation product from many oxidation pathways. These overall yields are similar to past estimates of isoprene's contribution to organic acid budgets (Millet et al., 2015; Müller et al., 2018).

   **Hydroperoxides:** Organic hydroperoxides serve as a $HO_x$ reservoir in the gas phase, can contribute to the oxidation of $SO_2$ to sulfate in aerosol and cloud water (Lind et al., 1987; Zhou and Lee, 1992), and are harmful to plants and human cells (Hewitt et al., 1990; Williams et al., 1983; Runge-Morris et al., 1989; Pöchl and Shiraiwa, 2015). We simulate an overall

molar yield of hydroperoxides from isoprene in excess of 50% globally (Table S3). As shown in Figure 11, the majority of this production (67%) consists of the first-generation hydroxy-hydroperoxide (ISOPOOH), with additional contributions from highly functionalized $C_4$ and $C_5$ compounds and from hydroperoxyacetone. Many of these are later-generation products of the initial ISOPOO + NO pathway, either through subsequent H-shifts or subsequent $RO_2$+$HO_2$ reactions; as a result, even in East China, where reaction with NO dominates the $RO_2$ fate, the molar yield of hydroperoxides from isoprene oxidation still

reaches 25%.

   **MVK and MACR:** MVK and MACR are formed in the first generation of isoprene oxidation via multiple pathways, with high production branching ratios from isoprene ozonolysis, H-shifts of $\beta$-ISOPOO isomers, and the reactions of ISOPOO with NO and $RO_2$. In the GEOS-Chem simulation with RCIM we find 28% and 16% global mean molar yields for MVK and MACR respectively. This represents a pronounced decrease in the relative yield of MACR compared to past mechanisms, largely

caused by the dynamic representation of the ISOPOO isomer distribution in RCIM and resulting titration of methacrolein-forming 4-OH-ISOPOO via rapid H-shift (see Section 3.3). The decreased relative importance of isoprene ozonolysis in RCIM, which generates 26% of global MACR, also contributes. This causes a sharp increase of up to 50% from past mechanisms in the simulated MVK/MACR ratio in diurnal-steady-state simulations to a range of 1.6-2.6 depending on $NO_x$ (see Figures S20 and S21 in the Supplement), in better agreement with observations (Greenberg et al., 1999; Karl et al., 2009; Wolfe

et al., 2016). (MVK+MACR)/isoprene and (MVK+MACR+ISOPOOH)/isoprene ratios, which are used as proxies for OH and photochemical age (Kuhn et al., 2007; Karl et al., 2009), remain considerably more consistent across mechanisms.

## 5.2   Organonitrates

RCIM includes important updates to the formation and fates of organonitrates through pressure- and temperature-dependent parameterizations of nitrate branching ratios and a new structure-activity relationship for calculating the formation of nitrates





from multifunctional peroxy radicals without measured yields (Wennberg et al., 2018). We also implement faster particle-phase hydrolysis of the 1-OH,2-ONO$_2$ isoprene hydroxynitrate (Darer et al., 2011; Hu et al., 2011; Rindelaub et al., 2014; Xu et al., 2015; Rindelaub et al., 2016), which decreases its ability to transport and recycle NO$_x$. Finally, the dynamic representation of the ISOPOO isomers causes lower production of methacrolein-derived nitrates, including MPAN, and higher production of the

hydrolysis-prone tertiary hydroxynitrate than previous mechanisms, as discussed in Section 3.3.

The effects of these updates are shown in Table 2 and Figures 9-11. Isoprene is found to contribute 20% of the tropospheric burden of peroxyacyl nitrates and 28% of non-peroxyacyl nitrates, with higher contributions in the Southern Hemisphere and local contributions up to 80% in regions of concurrent isoprene and NO$_x$ emissions. Among isoprene-derived PANs, peroxy-acetyl nitrate (PAN) dominates; we estimate that 19% of global PAN is derived from isoprene. This represents a considerably

smaller fraction than in the v11-02c mechanism (44%) and in Fischer et al. (2014) (37%), due primarily to lower yields of precursors; methylglyoxal yields are reduced due to higher losses of intermediates to deposition and SOA, and acetaldehyde is not produced from isoprene in RCIM. We simulate PAN/MPAN ratios between 10 and 20, similar to MCM and to observed ratios (Roberts et al., 2002; Cleary et al., 2007; Roberts et al., 2007; Jenkin et al., 2015). This is in contrast to the GEOS-Chem v11-02c mechanism, which found a large contribution from MPAN due to high methacrolein yields and a high formation rate

taken from Lin et al. (2013) (see Figure S23); the rate has since been revised down in GEOS-Chem v12.

GEOS-Chem with RCIM shows substantial daytime contributions from a number of non-PAN organonitrates (Figure 11), including first-generation C$_5$ hydroxynitrates as well as later-generation C$_4$ nitrates, C$_5$ tetrafunctionalized nitrates, and propanone nitrate, similar to those in GEOS-Chem v11-02c (Fisher et al., 2016) and MCM (see Section S5.3 of the Supplement). We simulate higher yields of first-generation hydroxynitrates than GEOS-Chem v11-02c but lower later-generation

yields, leading to an overall 14% decrease in non-PAN organonitrate production in RCIM. Nitrates derived from NO$_3$-initiated oxidation contribute substantially to nighttime burdens, but their relatively short lifetimes against photolysis and oxidation by OH mean they are rapidly lost in the day; for more on isoprene-NO$_3$ chemistry, see Section S4 of the Supplement.

In global simulations, as described in Section 4.2, the rapid hydrolysis of tertiary nitrates is an important sink of NO$_x$. Figure 8b shows the global fate of non-PAN isoprene-derived organonitrates in GEOS-Chem with RCIM. We find that only 20%

of these organonitrates recycle NO$_x$ via gas-phase oxidation, while 6% undergo deposition and 74% hydrolyze to inorganic nitrate. Organonitrate hydrolysis constitutes a NO$_x$ sink of 4.9 TgN a$^{-1}$ globally, or 10% of total NO$_x$ loss. Including organonitrate deposition and hydrogen abstraction from isoprene-derived VOCs by NO$_3$ to form HNO$_3$, the overall contribution of isoprene to global NO$_x$ loss reaches 15%. The increased hydrolysis rate also causes a 14% reduction in the production of C$_5$ tetrafunctional compounds which may contribute to SOA (see Table S2) because their organonitrate precursors are lost to

hydrolysis.

Over the Southeast United States, where isoprene nitrate chemistry has been extensively observed (Lee et al., 2016; Romer et al., 2016), we simulate that loss to hydrolysis is the fate of 69% of isoprene-derived nitrates annually, comprising 45% of the total regional NO$_x$ sink, while deposition and gas-phase NO$_x$ recycling contribute 9% and 22% respectively. Fisher et al. (2016) estimated a similar fraction of NO$_x$ recycling from isoprene nitrates of 23% over the Southeast United States in

summer, with a larger contribution from deposition (18%) and a smaller fraction lost to hydrolysis (59%). The average lifetime




of isoprene-derived nitrates in the region is 3.6 h in RCIM, more consistent with the observational estimate of 2-4 hours (Romer et al., 2016; Lee et al., 2016) than the simulated lifetimes of 0.48 days with GEOS-Chem v11-02c and 0.58 days in Horowitz et al. (2007).

## 5.3 SOA and its precursors

Figures 10-11 show the yields of major precursors of isoprene-derived SOA (iSOA) in RCIM, including IEPOX, highly functionalized $C_5$ compounds, HMML, other epoxides, and organonitrates, all of which are discussed in greater detail below. We find a global isoprene-derived SOA (iSOA) production of 61 TgC a$^{-1}$ (13% yield per carbon, Figure 8a) in GEOS-Chem using RCIM. IEPOX, tetrafunctional $C_5$ compounds, and organonitrates each contribute ~30% to this total (Figure 8c). If we simply consider the individual molecular weights of the iSOA precursors, we obtain a global iSOA source of 136 Tg SOA a$^{-1}$,

corresponding to a mass yield of 25% from isoprene, and an organic mass to organic carbon (OM/OC) ratio of 2.2 for iSOA, consistent with observations for highly oxidized SOA (Aiken et al., 2008). We assume this ratio in what follows, recognizing that subsequent aerosol-phase reactions not described here would modify it.

The 25% mass yield of SOA from isoprene simulated with RCIM is considerably higher than values commonly used in global models, e.g. the range of 0.9-6.8% in the models discussed in Carlton et al. (2009). The standard GEOS-Chem model

has two options for simulating iSOA: either a fixed mass yield of 3% (Kim et al., 2015) or an explicit representation coupled to the gas-phase mechanism (Marais et al., 2016). Marais et al. (2016) find from that explicit representation a 3.3% mass yield over the Southeast US in summer, with glyoxal and IEPOX dominating the formation in the high-NO and low-NO regimes respectively. In contrast, RCIM has a low yield of glyoxal from isoprene, as discussed above, and far larger contributions from other iSOA precursors. The Marais et al. (2016) GEOS-Chem mechanism has limited representation of iSOA formation from

tetrafunctional $C_5$ compounds and organic nitrates.

The ~3% mass yield of SOA from isoprene in GEOS-Chem was previously found to successfully account for organic aerosol observations over the US (Kim et al., 2015; Marais et al., 2016, 2017). A 25% mass yield would lead to a severe overestimate. However, these and other SOA observations in isoprene-dominated environments tend to be in high-NO conditions, where yields from IEPOX and tetrafunctionals are low (Figure 11). Organonitrates dominate iSOA formation in RCIM under high-

NO conditions but hydrolyze rapidly in the aerosol phase, and the organic moiety could further react and volatilize.

Recent work suggests that increased SOA yields from isoprene may be appropriate in global simulations, likely to be partially balanced by iSOA chemical sinks in order to reconcile with SOA observations. Global models tend to underestimate the atmospheric burden of organic aerosol (Volkamer et al., 2006; de Gouw and Jimenez, 2009; Tsigaridis et al., 2014). Chamber studies isolating specific isoprene oxidation pathways have measured SOA mass yields in excess of 15% (Liu et al., 2016;

Schwantes et al., 2019), while top-down (Hallquist et al., 2009; Heald et al., 2010; Spracklen et al., 2011) and mass-balance (Goldstein and Galbally, 2007) assessments of global SOA production consistently arrive at higher estimates than the models. Stadtler et al. (2018) found better agreement with the top-down assessments by implementing a new isoprene mechanism (including the major tetrafunctional $C_5$ compounds) into global simulations, resulting in a 33% global iSOA mass yield (16% per carbon). Hodzic et al. (2016) showed that discrepancies between observed SOA yields and modeled SOA budgets could be



reconciled by incorporating increased rates of SOA loss to deposition, photolysis, and heterogeneous reactions, balanced by SOA sources 3.9 times higher than the GEOS-Chem standard model.

**IEPOX:** The dominant contributor to iSOA worldwide (Marais et al., 2016; Stadtler et al., 2018), IEPOX is a second-generation oxidation product of isoprene via the ISOPOO + $HO_2$ reaction pathway. IEPOX can form in high yields of up to 75% from isoprene in $HO_2$-dominated conditions (Figure 10). In remote regions, these yields are strongly temperature-dependent due to competition from ISOPOO H-shift pathways. We estimate global IEPOX production to be 185 Tg a$^{-1}$, or a 20% molar yield from isoprene, similar to past estimates (Bates et al., 2014; St. Clair et al., 2015; Bates et al., 2016) and to the GEOS-Chem v11-02c mechanism (183 Tg a$^{-1}$). This results in 38 Tg a$^{-1}$ (20 TgC a$^{-1}$) of iSOA formation from IEPOX, slightly lower than a recent estimate by Stadtler et al. (2018). While the uptake parameterization of Marais et al. (2016) used here varies with particle acidity and sulfate content as seen in chamber studies and field observations (Gaston et al., 2014; Nguyen et al., 2014; Liao et al., 2015), it does not include the known effects of organic coatings and aerosol phase state (Riva et al., 2016; Zhang et al., 2018), which may also be important for the uptake of other precursors.

**$C_5$ tetrafunctional species:** RCIM includes eleven distinct $C_5$ tetrafunctional compounds with unique combinations of functional groups, each of which represents a variety of isomers. The global distribution of these compounds is shown in Figure 9, while their simulated daytime concentrations in diurnal-steady-state box models are shown in Figure 11 as a function of $NO_x$. $C_5$ dihydroxy-hydroperoxy-epoxides (IDHPE), formed in H-shift reactions following the addition of OH to ISOPOOH (D'Ambro et al., 2017), are estimated to contribute the bulk of the tetrafunctional compounds globally (54% of molar production) and under most $NO_x$ conditions. MCM and GEOS-Chem v11-02c predict similar yields of $C_5$ tetrafunctional species, but the relative contributions of individual species vary substantially between mechanisms (See Figures S22-23), and GEOS-Chem v11-02c only considered SOA formation from two such species (dihydroxy-dinitrates and "LVOC" produced in the reaction of ISOPOOH with OH). Because the rates of gas-phase oxidation, deposition, and aerosol uptake for these compounds are all poorly constrained, their contribution to iSOA remains highly uncertain.

While the individual yields of these compounds from isoprene may be small, their cumulative production (4.1% molar yield from isoprene globally) and relatively low volatility make them potentially substantial contributors to iSOA, and aerosol formation has been observed from these pathways in chamber experiments (Krechmer et al., 2015; D'Ambro et al., 2017). Quantitative descriptions of their contribution to SOA remain uncertain, but we estimate a global source of 46 Tg a$^{-1}$ (18 TgC a$^{-1}$) of iSOA from $C_5$ tetrafunctional species. IDHPE accounts for 51% of this total, with dihydroxy-dihydroperoxides and dihydroxy-hydroperoxy-carbonyls contributing over 10% each. This total carries high uncertainty, due both to the SOA uptake parameterization and the lack of constraints on other loss pathways of the $C_5$ tetrafunctional compounds, but is similar to a recent estimate by Stadtler et al. (2018) and highlights the importance of further investigations of this iSOA formation pathway.

**HMML**: Hydroxymethyl-methyl-$\alpha$-lactone, a product of methacrolein oxidation via MPAN, is considered a major iSOA precursor under high-$NO_x$ conditions (Nguyen et al., 2015a; Kjaergaard et al., 2012; Jiang et al., 2018). Its contribution to SOA is identified in ambient aerosol from its hydrolysis product, 2-methylglyceric acid (Edney et al., 2005; Szmigielski et al., 2007; Zhang et al., 2011). While HMML is better classified as a lactone, we include it with the epoxides in Figure 11 and Tables S1-S3, as it is thought to react similarly in aerosol (Jiang et al., 2018). Our mechanism shows only minor yields of HMML




under most conditions, up to a maximum of 2% molar yield from isoprene at extremely high NO (Figure 10), but it may contribute substantially to iSOA production locally; HMML production reaches 25% that of IEPOX in the $NO_x$-dominated conditions of East China (Table S3). RCIM results in similar production of HMML to MCM, but a lower yield than in GEOS-Chem v11-02c, largely due to lower MACR production and MPAN formation rates (see Section 5.2); as a result, we estimate

that HMML contributes only 0.18 Tg iSOA $a^{-1}$ (0.11 TgC $a^{-1}$) globally, much lower than the 1.7 Tg SOA $a^{-1}$ predicted in the v11-02c mechanism. However, a recent chamber study comparing iSOA yields to RCIM showed an underprediction of iSOA formation from the HMML pathway, suggesting that this global estimate may be too low (Schwantes et al., 2019).

**Other epoxides:** A new element of RCIM is the introduction of additional organic epoxide products. These include IDHPE, discussed above; $C_5$ carbonyl-hydroxy-epoxides (ICHE), produced from the reaction of IEPOX with OH and in the H-shifts of

$Z$-$\delta$-ISOPOO radicals (Bates et al., 2014; Wennberg et al., 2018); and two varieties of $C_5$ hydroxy-nitrooxy-epoxides, formed in the morning from the reactions of isoprene + $NO_3$ products with OH (Schwantes et al., 2015). The contributions of these epoxides relative to IEPOX in diurnal-steady-state simulations are shown in Figure 11; we find that they can comprise up to 20% of ambient epoxide concentrations under low-NO conditions. In global simulations, we find that non-IEPOX, non-IDHPE epoxides contribute 5.1 Tg $a^{-1}$ (2.6 TgC $a^{-1}$) of iSOA globally.

**Nitrates:** Multifunctional nitrates derived from both the ISOPOO + NO and isoprene + $NO_3$ pathways are also known to contribute to iSOA (Ng et al., 2008; Lee et al., 2014; Schwantes et al., 2019). In GEOS-Chem, nitrate hydrolysis results in irreversible iSOA formation; the higher organonitrate uptake and hydrolysis rates implemented in RCIM therefore result in a high iSOA formation from organonitrates of 49 Tg $a^{-1}$ (21 TgC $a^{-1}$). 6.9 Tg $a^{-1}$ (3.0 TgC $a^{-1}$) of this total comes from $C_5$ tetrafunctional compounds, and is already included in the amounts listed in that section above. Much of the rest comes

from $C_5$ difunctional compounds, which are expected to form alcohols (diols in the case of hydroxynitrates) following their particle-phase hydrolysis, many of which may be sufficiently volatile to partition back to the gas phase. The organonitrate iSOA formation simulated in GEOS-Chem is therefore likely an upper limit on the actual source of aerosol mass from organonitrates.

**Other compounds:** Additional known iSOA precursors include glyoxal and methylglyoxal. As discussed previously, we find that RCIM leads to low glyoxal yields relative to previous mechanisms; this results in a small estimated contribution of glyoxal

to global iSOA of 4.2 Tg $a^{-1}$ (1.7 TgC $a^{-1}$), 36% lower than in GEOS-Chem v11-02c and 58% lower than a recent estimate by Stadtler et al. (2018). In the Southeast United States, where Marais et al. (2016) found that glyoxal contributed about half as much iSOA as IEPOX, we instead find that production of iSOA from glyoxal is 10% of that from IEPOX, which matches the decreased glyoxal yield between the two mechanisms in the region. Locally, however, glyoxal can be important contributor to iSOA; we find that it contributes 23% of iSOA in East China. The production of methylglyoxal is much higher than that of

glyoxal (20% molar yield from isoprene globally), but due to its low SOA yield it contributes only 0.01 Tg iSOA $a^{-1}$ (McNeill et al., 2012). Finally, RCIM predicts a large molar yield of semivolatile highly oxidized $C_4$ compounds, including 51 Tg $a^{-1}$ of dihydroxy-carbonyls, 84 Tg $a^{-1}$ of hydroxy-dicarbonyls, and 56 Tg $a^{-1}$ of hydroxy-hydroperoxy-carbonyls, which may also contribute to iSOA formation; as with many other elements of the isoprene SOA formation scheme, further study is required to better constrain this pathway.





## 6 Further mechanism reduction

We use the results of the simulations described above to implement further simplifications to RCIM, and compile a "Mini" isoprene mechanism (Mini-CIM) for use in chemical transport modeling where computational cost is a concern. The speciation of highly functionalized isoprene oxidation products with low individual yields in RCIM goes beyond many measurement

capabilities and the needs of most atmospheric model applications. We therefore combine and remove many such products, with an aim toward maintaining the effects of isoprene on OH, $NO_x$, ozone, SOA precursors, readily measured organic products, and organonitrates as shown in Sections 3-5.

  In Mini-CIM, we create lumped species from isoprene oxidation products that meet two criteria: (1) $< 0.1\%$ molar yield from isoprene globally, and (2) $< 1\%$ molar yield from isoprene in each of the Amazon, Southeast United States, and East

China. These products are then lumped according to their number of carbon atoms (to conserve carbon) and similarity of lifetimes and functional groups. To maintain the effects of isoprene oxidation on $NO_x$ transport and removal, we prioritize lumping of functional groups by nitrate content. For example, all $C_5$ dinitrate compounds are lumped into a single species, while $C_5$ tetrafuntional mononitrates are lumped into two categories (those with and without an aldehyde, which substantially shortens the compounds' lifetimes). In addition to lumping species that meet the low-yield criteria, we remove peroxy radicals

that have recently been shown to undergo rapid H-shifts (Möller et al., 2019) and replace them with the products of those H-shifts. We further combine five pairs of isomeric species that exceed the molar yield thresholds but have identical loss rates and are predominantly produced concurrently, which means that their lumping has no effect on the species' lifetimes and minimal effect on product distributions. A detailed list of the simplifications made in the Mini mechanism can be found in the Supporting Information, along with a list of the excluded species and their global and regional molar yields (Table S7).

Global simulations with Mini-CIM exhibit only minimal differences from simulations with RCIM in the outcomes described in Sections 3-5. Table S6 shows the effects of these simplifications on the simulated global and regional production and burden of tropospheric radicals, ozone, SOA, and organic products. The tropospheric methane lifetime increases by only 0.1% from RCIM to Mini-CIM. Changes in annual average $HO_x$, $NO_x$, ozone, CO, and formaldehyde between the two mechanisms are all below 0.2% globally, and regional differences are only minimally larger. Changes in PANs, epoxides, and SOA are

below 0.5% globally and regionally, while $C_2$-$C_5$ nitrates and hydroperoxides exhibit similarly small global changes but some regional differences of up to 4.2% in areas with low absolute loadings.

  Whereas RCIM originally compiled in Wennberg et al. (2018) includes 148 organic species and 412 reactions, the new Mini-CIM contains 108 organic species involved in 345 reactions, which is comparable to the current mechanism in GEOS-Chem v11-02c (106 organic species involved in 335 reactions). We recommend the use of Mini-CIM in atmospheric models

except when more detailed speciation of highly functionalized, low-yield isoprene oxidation products is required for model-measurement comparisons. A complete listing of the species and reactions in Mini-CIM can be found in KPP format in the online repository with the original mechanisms (DOI 10.7907/Z9S75DHB).





## 7 Conclusions

We have presented a detailed analysis of the Reduced Caltech Isoprene Mechanism (RCIM), a new isoprene oxidation mechanism based on a recently developed explicit scheme (Wennberg et al., 2018), to examine its atmospheric implications for $HO_x$ and $NO_x$ radicals, ozone, organic products, and secondary organic aerosol (SOA) formation. We used for that purpose a

combination of box models and the GEOS-Chem global chemical transport model, and compared RCIM to the explicit MCM v3.3.1 and to the previous v11-02c version of the GEOS-Chem isoprene mechanism.

RCIM estimates a higher fraction of isoprene reacting with OH globally (88%) than past mechanisms. Of the fraction that reacts with OH to form hydroxy-peroxy radicals (ISOPOO), the dominant atmospheric fate is reaction with $HO_2$, while over 20% of ISOPOO radicals undergo H-shifts to regenerate $HO_x$. The dynamic system of ISOPOO isomers, and the differences

in H-shift rates between isomers, has important consequences for subsequent product formation. We show that the depletion of 4-OH ISOPOO due to its rapid H-shift leads to higher MVK/MACR ratios, higher tertiary nitrate production, and lower MPAN production than is simulated by mechanisms that do not treat the 1-OH and 4-OH ISOPOO systems separately.

The global effects of isoprene chemistry on radical families and ozone are similar in RCIM to past mechanisms, with notable regional differences. We find that isoprene is responsible for an 11% reduction in OH averaged over the troposphere,

causing a 12% increase in the tropospheric lifetime of methane. Depletion of OH under low-NO conditions is much less than in previous mechanisms because of $HO_x$ recycling from H-shift pathways. Isoprene oxidation results in a 6.5% increase in mean tropospheric $HO_2$ and a 4.2% decrease in $NO_x$. It increases tropospheric ozone by 1.9 ppbv globally but depresses ozone by up to 3.4 ppbv over tropical forests.

Mass conservation in RCIM enables a detailed accounting of the atmospheric fate of isoprene-derived carbon and the yields

of oxidation products. We find globally that 50% of isoprene is oxidized to $CO_2$ in the gas phase, 76% of which proceeds via CO and 44% via formaldehyde. Another 37% of isoprene-derived carbon is lost to organic deposition, while 13% forms SOA. For both formaldehyde and glyoxal, RCIM results in higher yields under low-NO conditions than previous mechanisms. However, deposition and aerosol uptake of isoprene oxidation intermediates greatly depresses the glyoxal yield relative to previous mechanisms.

The largest changes in RCIM relative to previous mechanisms are for organonitrates and SOA. We find that isoprene contributes 20% of the tropospheric burden of peroxyacyl nitrates and 28% of non-peroxyacyl nitrates, lower than in previous mechanisms. The implementation of fast tertiary nitrate hydrolysis leads to a $NO_x$ sink of 4.9 TgN $a^{-1}$ globally, or 10% of total $NO_x$ loss. Only 20% of isoprene-derived organonitrates (excluding peroxyacylnitrates) chemically recycle $NO_x$. We estimate the total global source of SOA from isoprene to be 61 TgC $a^{-1}$ (136 Tg $a^{-1}$), with approximately equal contributions

from IEPOX, organonitrates, and highly functionalized $C_5$ compounds. This 13% SOA yield per carbon is much higher than in previous global models, due primarily to our inclusion of additional precursors, but is similar to a recent estimate by Stadtler et al. (2018). Such high yields imply that SOA produced from isoprene cannot be regarded as chemically inert, and must further react in the aerosol phase to generate volatile products. This aerosol-phase chemistry is not yet included in RCIM and is a topic for further research.





Finally, we compiled a Mini-CIM mechanism that makes further simplifications to RCIM to decrease the computational burden of simulating isoprene chemistry. Mini-CIM has 108 species and 345 reactions, comparable in size to previous mechanisms implemented in GEOS-Chem while remaining closely consistent with the original mechanism of Wennberg et al. (2018). Global simulations with Mini-CIM exhibit minimal deviations from RCIM for atmospherically relevant applications.

5   *Code and data availability.*   The RCIM and Mini-CIM mechanisms used here are available online (DOI 10.7907/Z9S75DHB), along with the KPP code for conducting box model simulations and the model output discussed in this manuscript. MCM (http://mcm.leeds.ac.uk/MCM/) and GEOS-Chem (http://geos-chem.org) are both available online for public use.

*Author contributions.*   K. H. B. designed and carried out the simulations described herein, and prepared the manuscript with substantial assistance from D. J. J.

10   *Competing interests.*   The authors declare that they have no conflict of interest.

*Acknowledgements.*   K. H. B. acknowledges the support of the Harvard University Center for the Environment and the National Oceanic and Atmospheric Administration's Climate and Global Change Fellowship Programs. D. J. J. was supported by the US National Science Foundation Atmospheric Chemistry Program.





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


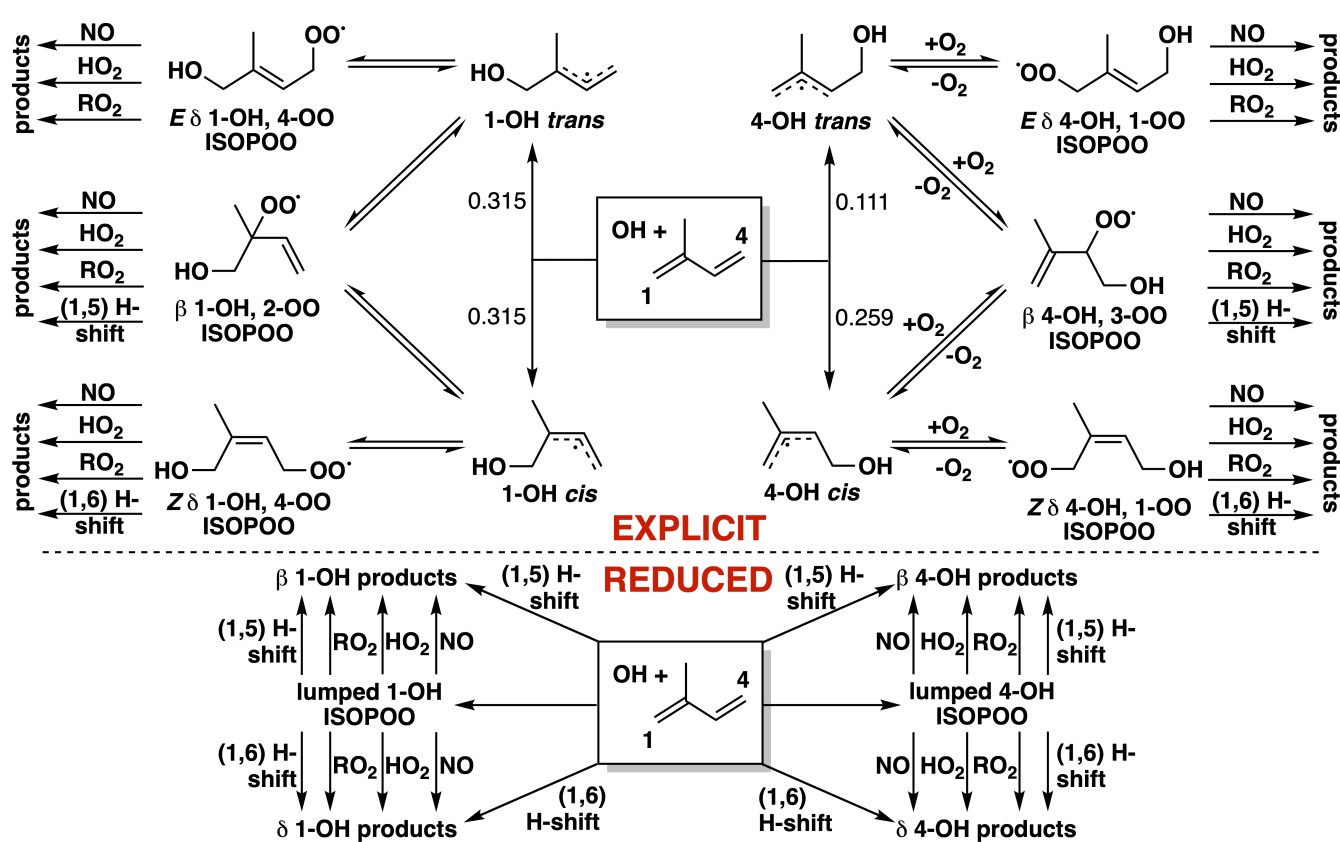

**Figure 1.** Fate of the allylic and peroxy radicals produced from the reaction of isoprene with OH in the presence of $O_2$. The explicit Wennberg et al. (2018) scheme is on top and the Reduced Caltech Isoprene Mechanism (RCIM) scheme is on bottom.





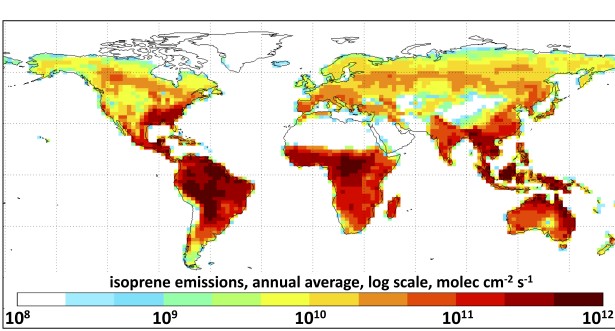

**Figure 2.** Annually averaged MEGAN v2.1 isoprene emissions for July 2014 – June 2015 as implemented in GEOS-Chem at $2° \times 2.5°$ horizontal resolution.



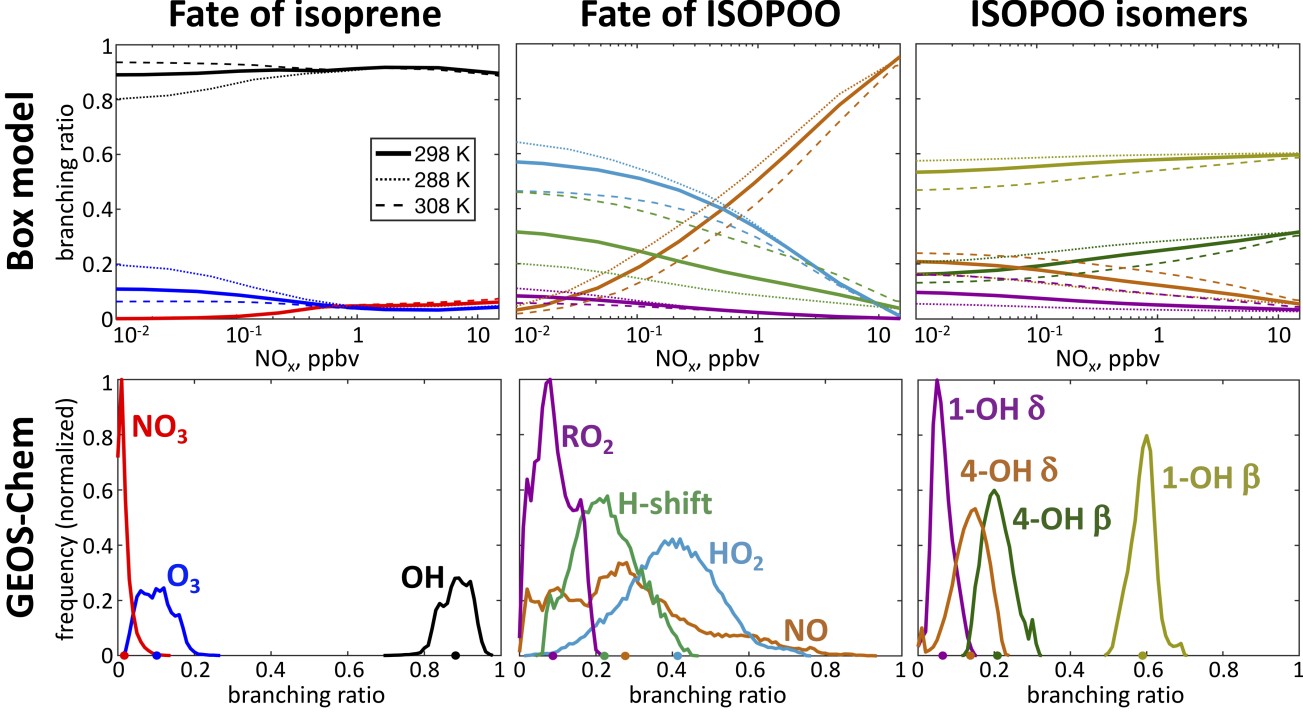

**Figure 3.** Fate of isoprene and isoprene hydroxy-peroxy radicals (ISOPOO) in the Reduced Caltech Isoprene Mechanism (RCIM). The figure shows isoprene oxidation pathway branching ratios in a diurnal-steady-state box model for clear-sky equatorial conditions (top) as a function of daytime mean $NO_x$ and temperature, and the global spatial frequency distribution of annual mean branching ratios in GEOS-Chem (bottom) at $2° \times 2.5°$ horizontal resolution. The GEOS-Chem frequency distributions are weighted by the amount of isoprene reacting in each grid box. Dots on the x axis indicate the global annual total reacting via each pathway. The distributions of ISOPOO isomers are weighted by their subsequent reactivity.





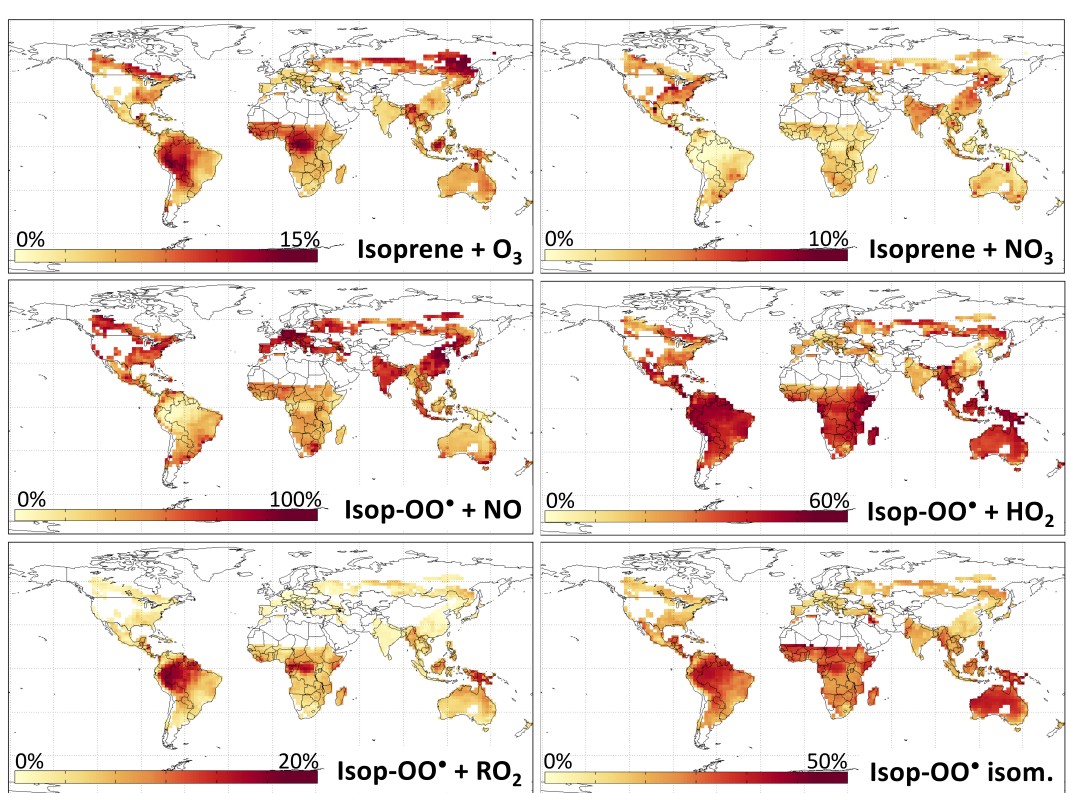

**Figure 4.** Percent of isoprene and the first-generation hydroperoxy radicals (ISOPOO) reacting via each pathway. Values are annual averages from the Reduced Caltech Isoprene Mechanism (RCIM) as implemented in GEOS-Chem and for the bottom 1 km of the troposphere. Grid boxes with an average isoprene oxidation rate of $< 1 \times 10^6$ molecules cm$^{-3}$ s$^{-1}$ are excluded. Note the different scales for each panel.



**Table 1.** Branching ratios (%) of isoprene oxidation pathways.[a]

| Pathway | | RCIM, global | v11-02c, global | Literature, global | RCIM, SE USA | RCIM, Amazon | RCIM, E China |
|---|---|---|---|---|---|---|---|
| Isoprene + | OH | 88 | 83 | $85^b$, $84^c$, $80^d$ | 85 | 86 | 91 |
| | $O_3$ | 10 | 15 | $9^b$, $11^c$, $15^d$ | 11 | 13 | 4.5 |
| | $NO_3$ | 1.7 | 2.3 | $5^c$, $5^b$, $5^d$, $6\text{-}7^{e,f}$ | 4.2 | 0.2 | 5.1 |
| ISOPOO + | $HO_2$ | 41 | 42 | $53.5^g$ | 31 | 45 | 14 |
| | NO | 28 | 31 | $33.5^g$ | 46 | 6.4 | 73 |
| | $RO_2$ | 8.8 | 13 | | 5.1 | 15 | 1.4 |
| | H-shift | 22 | 14 | $20^b$, $9.6^g$, $30^h$ | 18 | 33 | 11 |
| ISOPOO isomer[i] | *E/Z*-1-OH-$\delta$ | 6.5 | $2.4^j$ | $16^k$ | 5.5 | 10 | 4.1 |
| | 1-OH-$\beta$ | 59 | $51^j$ | $44^k$ | 59 | 55 | 61 |
| | *E/Z*-4-OH-$\delta$ | 14 | $18^j$ | $15^k$ | 12 | 19 | 7.9 |
| | 4-OH-$\beta$ | 21 | $28^j$ | $25^k$ | 23 | 16 | 27 |

[a] Annual totals. Percentage values from the Reduced Caltech Isoprene Mechanism (RCIM) implemented in GEOS-Chem are compared to the standard GEOS-Chem v11-02c mechanism and to literature values. Regional domains are defined in the text; SE USA = Southeast United States. [b] Müller et al. (2018). [c] Taraborrelli et al. (2009). [d] Pfister et al. (2008). [e] Horowitz et al. (2007). [f] Ng et al. (2008). [g] Crounse et al. (2011). [h] Peeters et al. (2014). [i] Values represent the reacted fractions, which govern product formation. [j] Inferred from product distribution. [k] Paulot et al. (2009a).



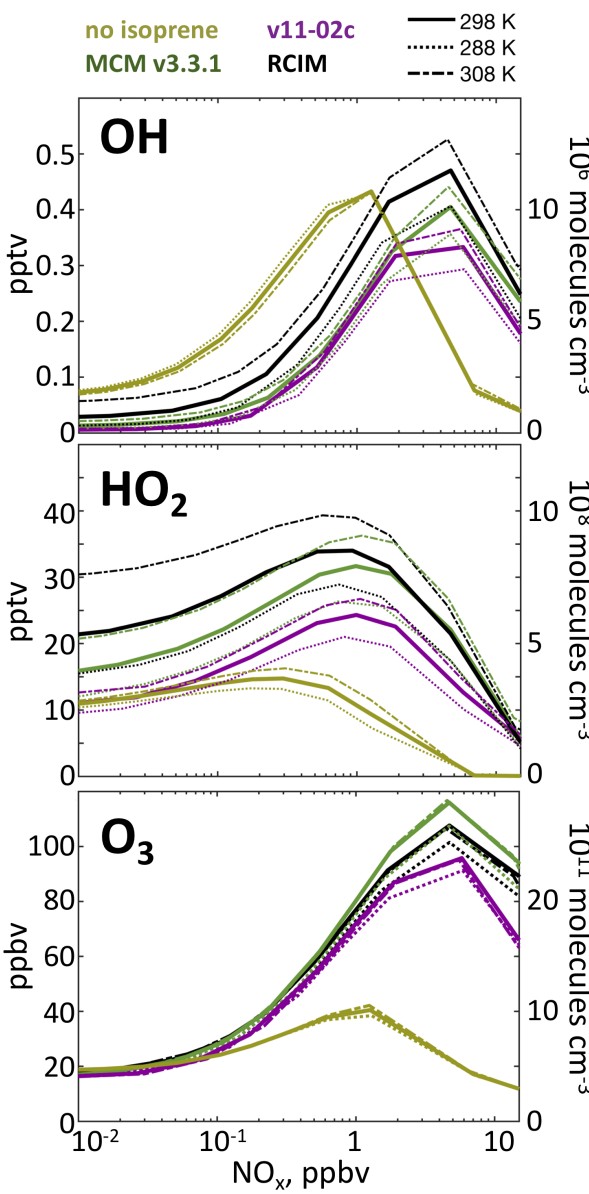

**Figure 5.** Effects of isoprene oxidation on OH, $HO_2$, and ozone concentrations in a diurnal-steady-state box model for clear-sky equatorial conditions as a function of $NO_x$ and temperature. The RCIM, MCM, and GEOS-Chem v11-02c mechanisms are compared.



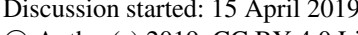



**Figure 6.** $HO_x$ production from H-shift chemistry in the Reduced Caltech Isoprene Mechanism (RCIM). The figure shows $HO_x$-generating pathways following the 1,6-H-shift of $Z$-$\delta$-1,4-ISOPOO. H-shift reactions are shown as dashed arrows, stable (closed-shell) products are shown in blue, and $HO_x$ production is shown in red. In RCIM, $C_4$-dihydroperoxy-aldehydes (top-right) are assumed to photolyze rapidly, resulting in a first-generation $HO_x$ recycling yield of 2.2 (1.5 OH + 0.7 $HO_2$) produced per ISOPOO H-shift.





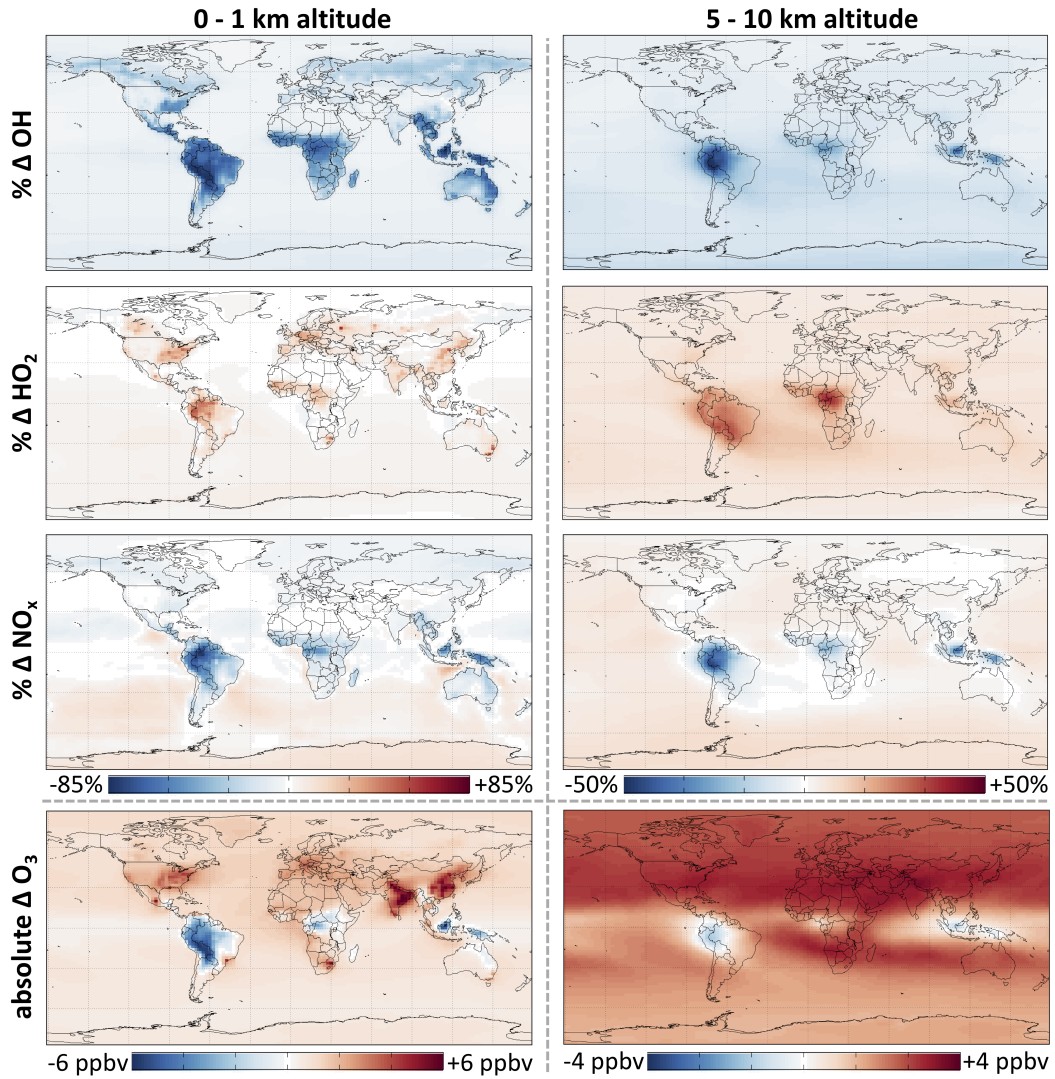

**Figure 7.** Effects of RCIM isoprene chemistry on OH, $HO_2$, $NO_x$, and ozone concentrations. The figure shows annual mean differences in GEOS-Chem simulations with versus without isoprene emissions.

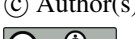



**Table 2.** Percent changes in tropospheric concentrations due to isoprene.[a]

| Species | Global | 0-1 km, global | 5-10 km, global | SE USA[b] | Amazon[b] | E China[b] |
|---|---|---|---|---|---|---|
| OH | -11 | -15 | -11 | -49 | -69 | -14 |
| $HO_2$ | 6.5 | 4.0 | 8.4 | 28 | 31 | 17 |
| $NO_x$ | -4.2 | -4.9 | -1.5 | -9.7 | -43 | -3.6 |
| $O_3$[c] | 4.2 (1.9) | 3.6 (0.9) | 4.1 (2.2) | 7.2 (3.0) | -22 (-3.4) | 9.1 (5.3) |
| CO | 30 | 25 | 32 | 27 | 60 | 7.5 |
| HCHO | 22 | 38 | 1.9 | 180 | 340 | 33 |
| PANs | 25 | 16 | 29 | 65 | 3.8 | 68 |
| Organonitrates[d] | 39 | 90 | 18 | 240 | 86 | 22 |

[a] Annual mean differences between GEOS-Chem simulations with and without isoprene emissions. Isoprene chemistry uses RCIM.

[b] Regional results are for 0-1 km altitude; see Section 2.2 for precise geographic definitions; SE USA = Southeast United States.

[c] Numbers in parentheses are annual mean absolute changes in ppbv. [d] Not including PANs.



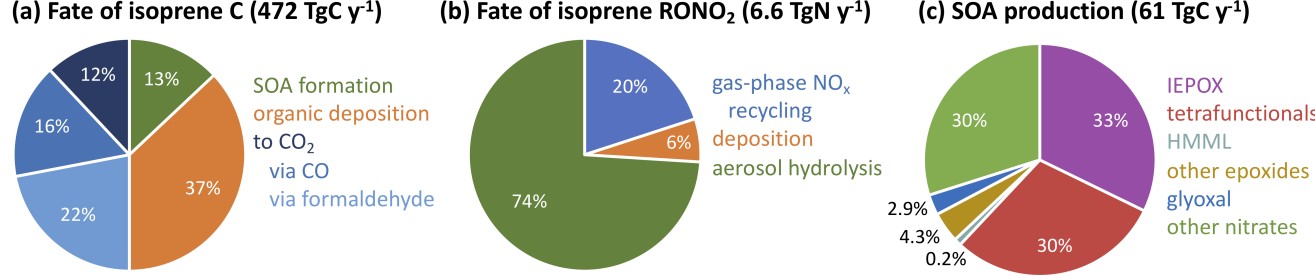

**Figure 8.** Accounting in the Reduced Caltech Isoprene Mechanism (RCIM) of (a) isoprene carbon, (b) the fate of isoprene-derived organon-itrates (not including PANs), and (c) contributions to isoprene-derived SOA. Values are global annual means from RCIM implemented in GEOS-Chem.





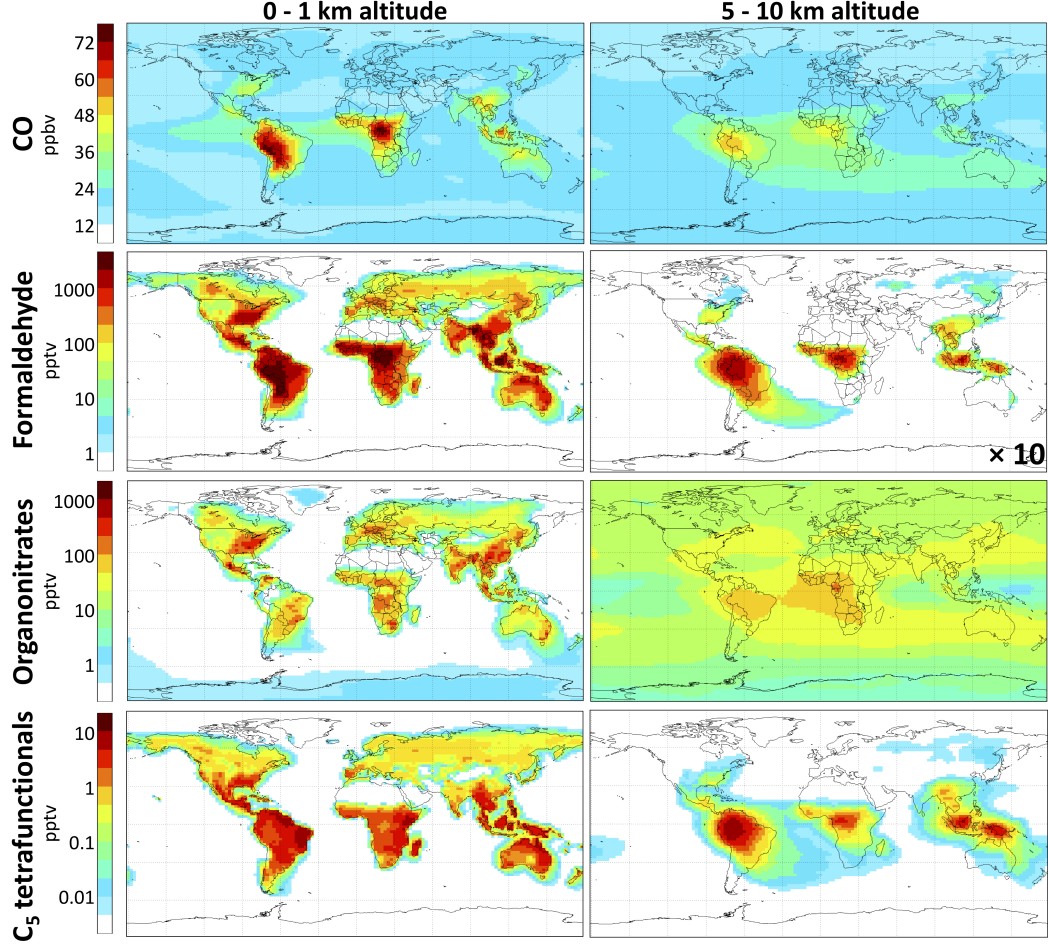

**Figure 9.** Contributions of isoprene to concentrations of CO, formaldehyde, organonitrates (including peroxyacyl nitrates), and $C_5$ tetrafunctional compounds using RCIM. Values are annual averages calculated as the differences between GEOS-Chem simulations with and without isoprene emissions. Color scale is linear for CO and logarithmic for other species.





**Figure 10.** Yields of organic products from isoprene + OH oxidation as a function of NO and $HO_2$. Results are from fixed-radical box model simulations with RCIM, run at 25 °C for clear-sky equatorial radiation at solar noon and an ozone column of 350 DU. The fixed-radical box model does not account for deposition or aerosol uptake. Contours are evenly spaced on a linear scale between the percent bounds listed on each plot. HPALDs ≡ $C_5$ hydroperoxy-aldehydes; ISOPOOH ≡ $C_5$ hydroxy-hydroperoxides; HMML ≡ hydroxymethyl-methyl-$\alpha$-lactone; MVK ≡ methyl vinyl ketone; MACR ≡ methacrolein.





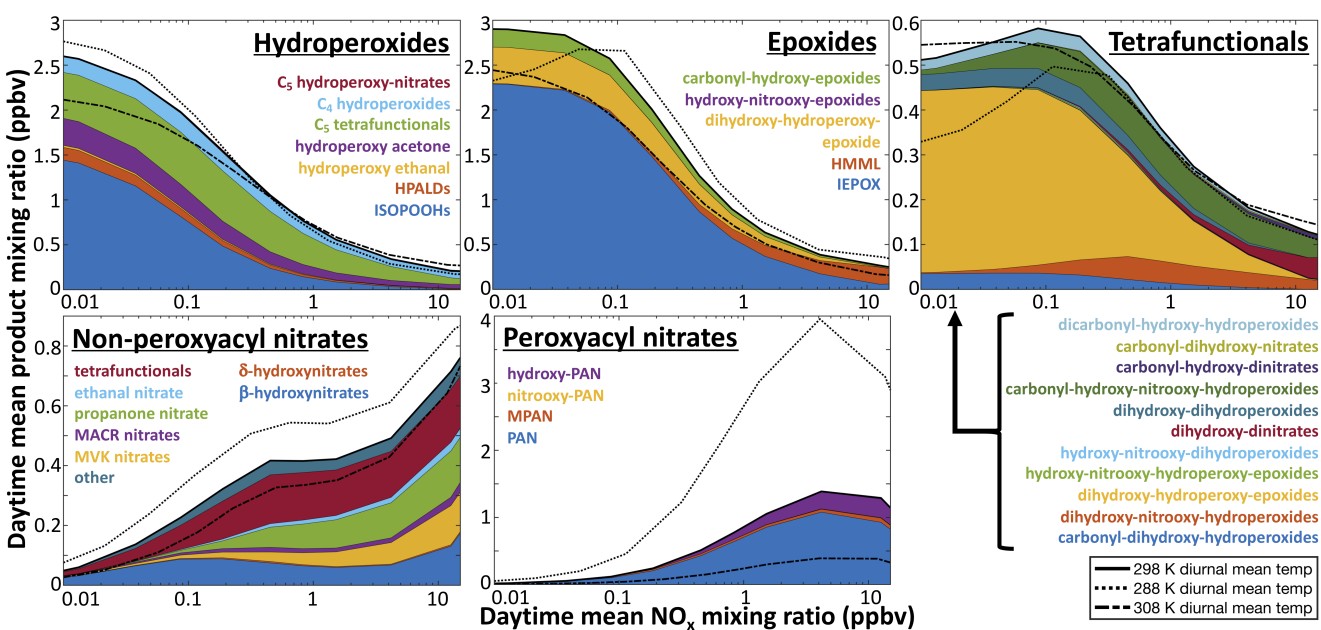

**Figure 11.** Daytime average concentrations of isoprene oxidation products as a function of $NO_x$. Results are from diurnal-steady-state box model simulations for equatorial conditions using RCIM. Y axis scales vary between panels.