# Peer review of "A new model mechanism for atmospheric oxidation of isoprene: global effects on oxidants, nitrogen oxides, organic products, and secondary organic aerosol"

_Atmospheric Chemistry and Physics, 2019_

## Referee Comment (RC1) · Anonymous Referee #1 · 17 May 2019

General comments

This paper presents a detailed assessment of the performance of the Reduced Caltech Isoprene Mechanism (RCIM) of Wennberg et al. (2018) in a series of models (box and GEOS-Chem global 3D), and uses it as a reference to produce a further condensed isoprene mechanism (the Mini-Caltech Isoprene Mechanism, Mini-CIM). Although still containing 108 species participating in 345 reactions, Mini-CIM is comparable in size to the existing GEOS-Chem isoprene mechanism (in GEOS-Chem v11-02c). The comparative performances of the reduced schemes in GEOS-Chem are presented and

discussed, with particular reference to the impacts of the updates on HOx recycling, NOx recycling and the formation of oxygenated products, including SOA precursors.

The performance of RCIM is also compared in detail with the performances of the isoprene chemistry in the Master Chemical Mechanism (MCM v3.3.1) and GEOS-Chem v11-02c in box model simulations. Similarities and differences are discussed in some detail, and related to specific pathways within the oxidation schemes.

This is a comprehensive and informative piece of work. It is important that the treatment of isoprene chemistry in global models reflects the significant developments in understanding that have occurred in recent years. Papers such as this, which aim to transfer the detailed understanding into reduced mechanisms in a transparent way, are therefore essential. This is therefore an important paper, which is appropriate for, and worthy of publication in, ACP. Below are a number of comments which the authors should consider in producing a revised version of the manuscript.

Specific comments

1) RCIM is a condensed version of the comprehensive (explicit) mechanism presented by Wennberg et al. (2018). In a few places, I found that I wanted a little more explanation of the underlying reasons for presented differences in performance of RCIM compared with the detailed MCM scheme, i.e. (i) are they due to different fundamental assumptions in MCM and the Wennberg et al. (2018) comprehensive mechanism; or (ii) do they result from simplifications made in RCIM compared with the Wennberg et al. (2018) comprehensive mechanism? If the former, are the differences because there is currently no experimental information to base the chemistry on, and different assumptions have been made? I think such clarification is necessary, because the differences are being presented as an improvement on former mechanisms (whether intentionally or unintentionally) and it is important for the reader to understand their origin. If based on provisional assumptions or conjecture, highlighting this can help guide future research effort.

Some specific examples are as follows (although I probably could list others):

a) Page 11, glyoxal section: The peak production of glyoxal at low [NO] from RCIM is explained by "….contributions from ISOPOO H-shifts and the degradation of IEPOX", and contrasted with that from MCM. Inspecting Figs. 3 and 8 of Jenkin et al. (2015), it appears that MCM does have routes to glyoxal (and methyl glyoxal) from both ISOPOO 1,6 H-shifts and the degradation of IEPOX, with this chemistry informed by mechanistic information presented by Peeters et al. (2014) and Bates et al. (2014). However, they rely on RO2 to RO conversion and therefore require reaction of RO2 with NO, NO3 or RO2. At low [NO] these processes are generally outrun by competing 1,4 formyl H atom shifts or reaction with HO2.

Wennberg et al. (2018) assume very high propagating channel branching ratios for the reactions of HO2 with RO2 radicals formed from OH + IEPOX, and those formed following the 1,6 ISOPOO H-shifts. These reactions provide additional RO2 to RO conversion routes at low [NO], with associated formation of glyoxal (and methyl glyoxal) and OH. This assumption may or may not prove to be correct, but at present there is no evidence from elementary kinetics studies of RO2 + HO2 reactions to support propagating branching ratios as high as some of those applied.

In the specific cases of the mechanisms following the ISOPOO 1,6 H-shifts, major products in both RCIM and MCM are dihydroperoxy carbonyl products, di-HPCARBs (called DHP-MVK and DHP-MACR by Wennberg et al., 2018), formed from 1,4 formyl H atom shift reactions of intermediate RO2. According to Fig. S4 of Wennberg et al. (2018), rapid exclusive photolysis of di-HPCARBs is assumed (instantaneous in RCIM) leading significantly to methylglyoxal formation, and some glyoxal formation (via HOOCH2CHO) in conjunction with substantial OH formation. Wennberg et al. (2018) indicate that "No experimental evidence exists to constrain these rates and products, so the mechanisms shown here are strictly conjectural". Because the di-HPCARBs do not contain a conjugated C=C double bond, it is likely that their photolysis is not exceptionally rapid, and MCM logically represents competitive loss by reaction with

OH. This reduces the yield, and delays formation, of methylglyoxal and glyoxal.

It therefore appears that at least some of the higher formation of glyoxal in RCIM at low [NO] results from assumptions that differ from those in MCM, which may or may not be correct. This therefore highlights areas of uncertainty in understanding, where more information is required, and this should probably be made clearer.

b) Page 8, from line 21: The higher concentrations of OH at low [NO] in RCIM compared with MCM are attributed to "... updated H-shift chemistry in RCIM, which efficiently recycles HOx..." with the ISOPOO 1,6 H-shifts producing "... 2.2 equivalents of HOx (1.5 OH + 0.7 HO2) concurrently with the first generation of stable products".

This is therefore related to the processes summarized above for glyoxal and methylglyoxal, in particular the assumed instantaneous photolysis of the major di-HPCARB products producing substantial OH and some HO2 (Fig. S4 of Wennberg et al., 2018). The competitive removal of di-HPCARBs by reaction with OH in MCM clearly leads to a significant difference in the HOx yield from this branch of the chemistry. As indicated above, Wennberg et al. (2018) also generally assume very high propagating channel branching ratios for the reactions of HO2 with RO2 radicals, which also helps to sustain HOx radical concentrations at low [NO].

It should probably be made clearer that the higher OH at low [NO] in RCIM results at least partly from the assumed choice of processes that maximize HOx regeneration, rather than from recent advances in understanding that are reported in Wennberg et al. (2018) and not considered in earlier mechanisms.

2) Page 5: I generally applaud the use of several modeling approaches, as they can each have advantages and disadvantages. However, the use of the "fixed radical box modeling" method would benefit from further justification of why it is of value. At present, it seems to be justified by the following sentence:

"This method serves to remove most nonlinearities and feedbacks inherent in the isoprene oxidation mechanism, so as to isolate the effects of the radicals on the oxidation pathways."

Surely, the non-linearities and feedbacks (i.e. on HOx and NOx) exemplify the major differences between the mechanisms and contribute to the "effects of the radicals on the oxidation pathways" in the different mechanisms. While I understand that you can look at the OH-initiated oxidation alone (i.e., without O3- and NO3-initiated oxidation), it is quite difficult to understand how heavily constraining the system provides reliable information on comparative mechanism performance.

Page 7, line 21: Related to above, > 10 ppb NO is stated to be "not of general atmospheric relevance". However, the fixed box model outputs in the SI present results and comparisons up to 100 ppb NO, with some of the largest differences occurring between 10 ppb and 100 ppb.

Minor comments

Page 1 line 20: Because simulated [OH] varies by an order of magnitude (e.g. Fig. 5), probably should give corresponding [OH] for the stated lifetime.

Page 2, line 23: I think "Heinz Becker" should simply be "Becker" (i.e. his first name is Karl-Heinz). Similarly in the reference list "Becker, K. H." rather than "Heinz Becker, K."

Page 3, line 1: I believe inclusion of Archibald et al. (2010) reference in this set of references is incorrect, because that study specifically did not consider hypothetical OH recycling mechanisms. In contrast, it systematically considered a series of explicit HOx recycling mechanisms based on reported experimental and theoretical data, and was one of the first (or possibly the first) to support and demonstrate the potential significance of the reversible O2 addition/peroxy radical isomerisation chemistry reported by Peeters et al. (2009), and subsequently characterized in detail by Wennberg and co-workers. The historical overview therefore needs some adjustment, and should also give more recognition to Peeters and co-workers for their pivotal role in moving the

understanding of isoprene chemistry forward.

Page 8, line 24: In relation to HOx production and recycling, the following statement is made about RCIM: "Assuming that photolysis is the dominant fate of the conjugated hydroperoxy-aldehydes (HPALDs) that make up 60% of the stable products, HOx production can increase . . ."

If I understand pp 3352/53 of Wennberg et al. (2018) correctly, the conjugated HPALDs actually only account for 25 % of the products following 1,6 H isomerization; with 15 % unconjugated HPALDs and the remainder other products (e.g. di-HPCARBs). This seems quite different from the stated 60 %. Looking at Fig. 6, it looks like the conjugated species make up 60 % of the total HPALDs, but are 24 % (i.e. 60 % of 40 %) of the full suite of products (with 16 % being the unconjugated species). If this is correct, I presume that the statement on Page 8, line 5 should specify "60% of the HPALDs" rather than "60% of the stable products".

Fig. S4 caption – I think "fun" should be "run".

Fig. S19 caption – I think "Jenkin et al. 2015" here is incorrect.
* * *

---

## Referee Comment (RC2) · Anonymous Referee #2 · 17 Jun 2019

Overview:

This paper presents an updated isoprene oxidation mechanism (RCIM) that is suitable for global modelling studies and has consequences for the burdens of several important trace species in the atmosphere; namely NOx, ozone, HOx, as well as the secondary organic aerosol (SOA). Given isoprene's importance in the troposphere, and the large amount of activity spent over the last decade elucidating its oxidation mechanism, this is clearly important work.

[Figure]

The paper starts with a brief description of the important changes to the mechanism relative to other mechanisms such as the MCM v3.3.1. These include new products from H-shift intramolecular reactions, new parameterisations for the nitrate yield from NO + RO2 reactions, explicit treatment of the tetra-functionalised compounds and more detailed NO3 chemistry. The effects of the new mechanism relative to other isoprene oxidation mechanisms are investigated using two box model studies and incorporation of the reduced mechanism into the GEOS-Chem global model.

While overall it is felt that this is a good paper representing a valuable addition to the literature, the reviewer has some concerns that should be addressed before the paper is published.

General comments:

In section 2.2, the three methods of analysing the RCIM are discussed. The first is a fixed radical box model where the concentrations of NO, HO2 are kept constant and the only oxidant present is OH. The second is a diurnal steady state box model which has varying radical concentrations and temperature, as well as varying emissions of isoprene and NO to simulate a tropical boundary layer. Finally, RCIM is incorporated into the global model GEOS-Chem. This is initialised for 18 months before being run for 1 year. The diurnal steady model is well explained with the daytime average value from the 8th day serving as the output. The GEOS-Chem run is also adequately explained. However, it is felt that more explanation about the analysis of the fixed radical box modelling is necessary. It is understood that the model is run until complete conversion to CO2 but it is not explained how the concentration for a particular species is calculated; is it the maximum value achieved by a species, the average concentration over a period of time or another metric?

It appears the main aim of the paper is to compare the effect of the new isoprene mechanism with older mechanisms. Therefore, it is felt that Figure 7, which compares global model results of the new isoprene mechanism with a no-isoprene scenario, is

much less relevant than Figure S17 in the SI which compares global model results between the new mechanism and the standard GEOS-Chem vn11.02 mechanism. The general effect of isoprene on NOx, O3 and OH is well known in the field. Fig S17 should replace current Figure 7 and the discussion in section 4 should focus more on the differences in global model output between vn11.02 and RCIM rather than RCIM vs. no isoprene. Furthermore, the no-isoprene scenarios plots in Figure 5 should be removed for clarity and more attention paid to the differences between the various mechanisms' outputs.

Continuing on this issue, with respect to the global model comparison section, it would be beneficial to see how the model output using RCIM and vn11.02 compare to observational data. In particular the significant predicted changes to OH, NOx, CO and HCHO over the Amazon and the CO change over much of the southern hemisphere should be compared to observational data if one is to have confidence in the use of RCIM.

The SOA yield is predicted to be significantly higher than previous models. The contribution to SOA from various species is explained. However, little detail is provided regarding the estimates of SOA production from each species aside from IEPOX. Specifically, the estimates of SOA from HMML, non-IEPOX non-IDHPE species, nitrates, glyoxal and the tetrafunctionalised species are not explained. The decision to treat the tetrafunctionalised species as LVOCs within the GEOS-Chem framework also warrants further discussion as the species span a wide range of volatilities.

---

## Author Comment (AC1) · 2 Jul 2019

**RESPONSE TO REVIEWERS**
21 June 2019
Kelvin H. Bates and Daniel J. Jacob

We sincerely appreciate the time and effort expended by the reviewers in helping us improve and refine this lengthy manuscript. We have responded to each of the reviewers' suggestions in detail below. In the following pages, the reviewer comments are shown in italics, while our responses are shown in plain text. Throughout the responses below and the attached revised manuscript, added text is highlighted in red, while removed text is greyed and crossed out. We hope these revisions have fully addressed the concerns of the reviewers.

**REVIEWER 1**

*I found that I wanted a little more explanation of the underlying reasons for presented differences in performance of RCIM compared with the detailed MCM scheme, i.e. (i) are they due to different fundamental assumptions in MCM and the Wennberg et al. (2018) comprehensive mechanism; or (ii) do they result from simplifications made in RCIM compared with the Wennberg et al. (2018) comprehensive mechanism? If the former, are the differences because there is currently no experimental information to base the chemistry on, and different assumptions have been made?*

Very few of the differences in performance presented herein are due to changes from the comprehensive (full, explicit) mechanism to the reduced (RCIM) mechanism. The Reduced mechanism was made concurrently with the full in Wennberg *et al.* (2018) and was designed to keep final product yields of known compounds the same. Where such products and reaction pathways weren't known, the authors sought to apply reasonable assumptions, extrapolations from similar compounds, and structure-activity relationships. Thus, most of the differences presented here are due to fundamental differences between the chemistry in RCIM and that in MCM and the GEOS-Chem v11-02c mechanisms, which we seek to describe along the way as we present the differences in model outcomes.

For the most part, the simplifications made from the full mechanism to the RCIM are detailed in Wennberg *et al.* (2018), and we avoid rehashing them here in this already length paper. To clarify these points, we have added the following to the first paragraph in the "Chemical mechanism" section (2.1): "RCIM is v4.1 of the "Reduced-plus" mechanism found in the Wennberg *et al.* (2018) mechanism repository (DOI 10.7907/Z9S75DHB). It includes the oxidation of isoprene by OH, ozone, and $NO_3$ and condenses the ensuing oxidation cascade for the practical range of atmospheric conditions. The mechanism includes 148 species and 412 reactions representing the complete isoprene oxidation cascade , in contrast to the 385 species and 810 reactions in the Wennberg *et al.* (2018) full explicit mechanism, which did not seek to provide loss processes for compounds without experimental constraints, and therefore did not represent a complete oxidation cascade (i.e. full conversion to $CO_2$). RCIM was compiled concurrently with the full explicit mechanism and was designed to

keep product yields of known compounds the same, with minimal simplifications beyond lumping of isomeric compounds with similar reaction pathways and removal of especially minor (<1% yield) pathways. Under atmospheric conditions, early-generation compound yields and mixing ratios in simulations with RCIM therefore closely track those of the full explicit mechanism. Major deviations occur only for later-generation compounds for which minimal experimental evidence exists to constrain reactive pathways, and for the proposed products of these reactions. For such compounds, the authors applied a self-consistent set of assumptions (Section 2 in Wennberg *et al.,* 2018) based on extrapolation from similar compounds and structure-activity relationships. While these assumptions were grounded in experimental evidence, they necessarily include high levels of uncertainty, which are discussed in greater detail in Section 8 of Wennberg *et al.* (2018)."

*a) Page 11, glyoxal section: The peak production of glyoxal at low [NO] from RCIM is explained by ". . ..contributions from ISOPOO H-shifts and the degradation of IEPOX", and contrasted with that from MCM. Inspecting Figs. 3 and 8 of Jenkin et al. (2015), it appears that MCM does have routes to glyoxal (and methyl glyoxal) from both ISOPOO1,6 H-shifts and the degradation of IEPOX, with this chemistry informed by mechanistic information presented by Peeters et al. (2014) and Bates et al. (2014). However, they rely on $RO_2$ to RO conversion and therefore require reaction of $RO_2$ with NO, $NO_3$ or $RO_2$. At low [NO] these processes are generally outrun by competing 1,4 formyl H atom shifts or reaction with $HO_2$. Wennberg et al. (2018) assume very high propagating channel branching ratios for the reactions of $HO_2$ with $RO_2$ radicals formed from OH + IEPOX, and those formed following the 1,6 ISOPOO H-shifts. These reactions provide additional $RO_2$ to RO conversion routes at low [NO], with associated formation of glyoxal (and methyl glyoxal) and OH. This assumption may or may not prove to be correct, but at present there is no evidence from elementary kinetics studies of $RO_2$ + $HO_2$ reactions to support propagating branching ratios as high as some of those applied.*

The discrepancies in glyoxal between GEOS-Chem v11-02c and RCIM are among the most striking in this manuscript, and they do therefore deserve particular attention, as the reviewer points out. Glyoxal yields were a point of strong disagreement between MCM and GEOS-Chem to begin with (fig S18-19); while MCM predicts higher glyoxal than RCIM under NO-dominated conditions, lower glyoxal under $HO_2$-dominated conditions, and about the same under isomerization-dominated conditions, GEOS-Chem predicts about the same under all conditions except isomerization-dominant, under which conditions it predicts much more glyoxal. This derived largely from the observational constraints presented in Chan Miller *et al.* (2017), who sought to reconcile high observed glyoxal in isomerization-dominated conditions by increasing second-generation production from the products of isomerization (HPALD, DHDC). While RCIM retains some second-generation formation from the non-HPALD products (namely via HPETHNL from the assumed rapid photolysis of the $C_4$ dihydroperoxy carbonyls) and later-generation formation (via glycolaldehyde and the $C_4$ tetrafunctional compounds), the yield through these pathways is greatly reduced from Chan Miller's work. As pointed out by the reviewer, deviations from MCM derive largely from the radical-propagating channels of $RO_2$ + $HO_2$, which are assumed to be greater in RCIM than in MCM.

In an already lengthy paper, we sought to avoid discussing specific points of uncertainty in the mechanism when such details are already covered in the initial mechanism description in Wennberg *et al.* (2018), but we agree that pointing out the conjectural pathways that prove particularly important to model outcomes of interest may be a beneficial way to direct future research to the most important remaining uncertainties, and will help the reader understand which aspects of these model outcomes and more or less constrained. As such, we have added a brief discussion of the uncertainties in the Wennberg *et al.* (2018) mechanism to Section 2.1 (described in the response to the previous reviewer comment; the radical propagating channels of $HO_2$ + $RO_2$ reactions were highlighted as one such uncertainty), and have revised the glyoxal subsection of Section 5.1 as follows:

"Glyoxal ($C_2H_2O_2$) is also measured by satellites (Vrekoussis *et al.,* 2009; Alvarado *et al.,* 2014; Chan Miller *et al.,* 2014), and different yields relative to formaldehyde can discriminate between emissions of different VOCs (Chan Miller *et al.,* 2016). Past mechanisms have provided differing estimates on which isoprene oxidation pathways produce the most glyoxal (Li *et al.,* 2016), and comparisons with field measurements show that glyoxal production is higher under low-NO conditions than most mechanisms predict (Li *et al.* 2016, Chan Miller *et al.*, 2017).  In contrast, glyoxal yields in MCM are highest at high NO. Our diurnal steady-state box model simulations show that the RCIM glyoxal/formaldehyde ratio remains in the 2-3% range over the ensemble of atmospheric conditions (see Figure S21), in line with field observations for isoprene-dominated environments (Kaiser *et al.,* 2015; Chan Miller *et al.,* 2017).

RCIM yields of glyoxal from isoprene peak at 10% under low-NO conditions (Figure 10), while glyoxal yields in MCM are highest under high-NO conditions, and yields in the GEOS-Chem v11-02c mechanism are even higher than RCIM under low-NO conditions (Figures S18-S19). Mechanistically, these differences primarily reflect changes in the contributions from two low-NO pathways in RCIM relative to MCM and v11-02c: the products of $Z$-$\delta$-ISOPOO H-shifts, and the reactions of IEPOX-derived peroxy radicals with $HO_2$. While both MCM and RCIM include moderate yields of glyoxal (largely via hydroperoxyethanal) from the $C_4$-dihydroperoxy-carbonyl products of $Z$-$\delta$-ISOPOO H-shifts, GEOS-Chem v11-02c incorporates much higher second-generation glyoxal yields from these H-shift pathways (primarily via HPALD and dihydroperoxy-dicarbonyl compounds), consistent with field observations (Chan Miller *et al.,* 2017). For the reactions of IEPOX-derived peroxy radicals with $HO_2$, both RCIM and GEOS-Chem v11-02c include moderate yields of glyoxal presumed to form in the radical-propagating reaction channel ($RO_2$ + $HO_2$ → RO + OH + $O_2$), as suggested in Bates *et al.* (2014) and implemented in Wennberg *et al.* (2018), while MCM includes no glyoxal formation under low-NO conditions from IEPOX-derived peroxy radicals. Both the atmospheric fates of $C_4$-dihydroperoxy-carbonyl compounds and the radical-propagating channels of non-acyl $RO_2$ + $HO_2$ reactions are poorly constrained (Wennberg *et al.,* 2018), and the glyoxal yields from these pathways therefore remain uncertain.

We find in GEOS-Chem that many glyoxal precursors (IEPOX, nitrates, and tetrafunctional $C_5$ compounds) are lost to aerosol or deposition before they can react in the gas phase, depressing the glyoxal yield relative to the box model simulations where aerosol/deposition effects are not included. This results in a global glyoxal yield from isoprene of 2% in GEOS-Chem with RCIM, only half that reported recently by Muller *et al.* (2019) and even lower than in some past simulations (Fu *et al.,* 2008; Myriokefalitakis *et al.,* 2008; Taraborrelli *et al.,* 2009). We find a reduction in global tropospheric glyoxal loading of 60% relative to the GEOS-Chem v11-02c mechanism. However, Miller *et al.* (2017) found good agreement between glyoxal simulated by GEOS-Chem v11-02c and aircraft observations in the Southeast United States. This suggests that RCIM may underestimate glyoxal yields from isoprene."

*Rapid exclusive photolysis of di-HPCARBs is assumed (instantaneous in RCIM) leading significantly to methylglyoxal formation, and some glyoxal formation (via HOOCH2CHO) in conjunction with substantial OH formation. Wennberg et al. (2018) indicate that "No experimental evidence exists to constrain these rates and products, so the mechanisms shown here are strictly conjectural". Because the di-HPCARBs do not contain a conjugated C=C double bond, it is likely that their photolysis is not exceptionally rapid, and MCM logically represents competitive loss by reaction with OH. This reduces the yield, and delays formation, of methylglyoxal and glyoxal. It therefore appears that at least some of the higher formation of glyoxal in RCIM at low [NO] results from assumptions that differ from those in MCM, which may or may not be correct. This therefore highlights areas of uncertainty in understanding, where more information is required, and this should probably be made clearer.*

The uncertainties in this pathway do indeed merit greater attention, and we thank the reviewer for the detailed assessment of this reaction channel. We highlight these uncertainties and their importance for both glyoxal production and OH regeneration in our responses to the previous reviewer comment (glyoxal, Section 5.1) and the next reviewer comment (OH, section 4.1).

*It should probably be made clearer that the higher OH at low [NO] in RCIM results at least partly from the assumed choice of processes that maximize $HO_x$ regeneration, rather than from recent advances in understanding that are reported in Wennberg et al. (2018) and not considered in earlier mechanisms.*

This process is indeed a major source of the increased OH in RCIM under low-NO conditions, and we thank the reviewer for drawing more attention to this uncertainty. As described above, we have included additional discussion of the relative certainty and uncertainty of specific aspects of the Wennberg *et al.* (2018) mechanism in Section 2.1, which touches on this point. We have also added the following to the "Effects on $HO_x$ radicals" section (4.1) to clarify the uncertainty surrounding the fate of the $C_4$ dihydroperoxy-carbonyl species: " The initial H-shift of the *Z-δ-4-OH-ISOPOO* radical (the dominant ISOPOO H-shift pathway) is highly temperature-dependent and regenerates one equivalent of $HO_x$ (0.6 OH + 0.4 $HO_2$) concurrently with the first generation of non-radical products. In RCIM, the $C_4$-dihydroperoxy-carbonyl compounds (top right of Figure 6) produced in this reaction are assumed to rapidly photolyze as postulated in

Wennberg *et al.* (2018), which produces an additional 1.2 $HO_x$ equivalents, for a total $HO_x$ regeneration of 2.2 equivalents (1.5 OH + 0.7 $HO_2$) from the 1,6 H-shifts of $Z$-$\delta$-ISOPOO isomers. Reaction with OH could possibly provide a competitive loss pathway for the $C_4$-dihydroperoxy-carbonyl compounds, which would result in lower net $HO_x$ production."

*The use of the "fixed radical box modeling" method would benefit from further justification of why it is of value. At present, it seems to be justified by the following sentence: "This method serves to remove most nonlinearities and feedbacks inherent in the isoprene oxidation mechanism, so as to isolate the effects of the radicals on the oxidation pathways." Surely, the non-linearities and feedbacks (i.e. on $HO_x$ and $NO_x$) exemplify the major differences between the mechanisms and contribute to the "effects of the radicals on the oxidation pathways" in the different mechanisms. While I understand that you can look at the OH-initiated oxidation alone (i.e., without $O_3$- and $NO_3$-initiated oxidation), it is quite difficult to understand how heavily constraining the system provides reliable information on comparative mechanism performance.*

Our goal in including the fixed-radical box models was to provide quantitative product yields, particularly of organic products, under specific ambient conditions; their use is intended primarily for the reader who has observed isoprene oxidation in a chamber or the atmosphere at quantified NO and $HO_2$ conditions to look up an expected yield from these plots. We acknowledge in the text that this is an unrealistic way to compare the effects of isoprene oxidation on oxidant cycling and overall (e.g. global) outcomes across mechanisms, but we still find utility in answering the question: "Under a given observed level of (e.g.) OH, NO, and $HO_2$, light, and temperature, what can I expect the yield of (e.g.) formaldehyde to be from OH-initiated isoprene oxidation in RCIM, and how does that compare to MCM and GC v11?" In an effort to clarify this point, we have expanded the sentence questioned by the reviewer to read: "This method serves to remove most nonlinearities and feedbacks inherent in the isoprene oxidation mechanism, so as to isolate the effects of the radicals on the oxidation pathways, and provides a quantitative reference of organic product yields from OH-initiated isoprene oxidation under fixed ambient conditions."

*> 10 ppb NO is stated to be "not of general atmospheric relevance". However, the fixed box model outputs in the SI present results and comparisons up to 100 ppb NO, with some of the largest differences occurring between10 ppb and 100 ppb.*

We believe these may still be a useful reference to some readers, particularly because while such conditions are not of general atmospheric relevance, they have been used in chamber experiments to constrain product yields. The comparisons in, e.g., Figure S14 therefore allow a reader with some specific curiosity about these conditions to identify the potential disparities in outcomes between the mechanisms compared here. We have updated the relevant sentence from "Here, the reduced model deviates substantially from reality and from the explicit model, but these conditions are rarely relevant in the atmosphere." to "Here RCIM deviates substantially from the explicit mechanism of Wennberg et al. (2018) and MCM (see Figure S14). These conditions are not of general atmospheric relevance, but may occur in chamber

experiments; for such applications, we recommend the use of a mechanism that resolves the full system of allylic and peroxy radicals (Figure 1)."

*Because simulated [OH] varies by an order of magnitude (e.g. Fig. 5), probably should give corresponding [OH] for the stated lifetime.*

We have added a parenthetical clarification to the relevant sentence: "($\tau_{OH}$ = 1.1 h for [OH] = 2.5 x $10^6$ molecules cm$^{-3}$ at *T* = 298 K)".

*I think "Heinz Becker" should simply be "Becker" (i.e. his first name is Karl-Heinz). Similarly in the reference list "Becker, K. H." rather than "Heinz Becker, K."*

This copyediting error has been fixed as suggested.

*I believe inclusion of Archibald et al. (2010) reference in this set of references is incorrect, because that study specifically did not consider hypothetical OH recycling mechanisms. In contrast, it systematically considered a series of explicit $HO_x$ recycling mechanisms based on reported experimental and theoretical data, and was one of the first (or possibly the first) to support and demonstrate the potential significance of the reversible $O_2$ addition peroxy radical isomerisation chemistry reported by Peeters et al. (2009), and subsequently characterized in detail by Wennberg and coworkers. The historical overview therefore needs some adjustment, and should also give more recognition to Peeters and co-workers for their pivotal role in moving the understanding of isoprene chemistry forward.*

Condensing the expansive and convoluted history of research on isoprene oxidation mechanisms into a brief overview was a challenge, and we thank the reviewer for pointing out this oversight. We have removed the reference to Arichibald *et al.* (2010) from the sentence describing hypothetical OH-recycling mechanisms, and have replaced the subsequent sentence ("These were later replaced with mechanistic OH-recycling pathways, including isoprene epoxydiol (IEPOX) formation (Paulot *et al.,* 2009b) and H-shift chemistry (Peeters *et al.,* 2009; Asatryan *et al.,* 2010; Crounse *et al.,* 2011). ") as follows: "These were later replaced with mechanistic OH-recycling pathways, including isoprene epoxydiol (IEPOX) formation (Paulot *et al.,* 2009b), radical propagation in reactions of $HO_2$ with acylperoxy radicals (Hasson *et al.*, 2004; Jenkin *et al.*, 2007; Dillon and Crowley, 2008), and H-shift isomerizations of the initial isoprene-hydroxy-peroxy radicals (Peeters *et al.,* 2009). Incorporation of these mechanistic OH-recycling pathways into models showed the latter pathway to be most important for sustaining elevated OH concentrations under low-NO conditions (Archibald *et al.*, 2010), and subsequent studies have identified and characterized additional OH-regenerating H-shift reactions throughout the isoprene oxidation mechanism (Peeters *et al.,* 2010; Crounse *et al.*, 2012; Crounse *et al.*, 2013; Peeters *et al.*, 2014; Jørgensen *et al.*, 2016; Wang *et al.,* 2018; Møller *et al.*, 2019)."

We have also expanded the sentence in the subsequent section (2.1) describing the dynamic system of hydroxy-isoprene-allylic and hydroxy-isoprene-peroxy (ISOPOO) radicals to

reflect the role of Petters and coworkers in elucidating this chemistry. The sentence "Addition of $O_2$ to allylic radicals under ambient conditions is in fact a reversible process, resulting in a dynamic system with differing initial (kinetic) and equilibrium radical distributions (Teng *et al.*, 2017)" now reads "Addition of $O_2$ to allylic radicals under ambient conditions is in fact a reversible process, resulting in a dynamic system with differing initial (kinetic) and equilibrium radical distributions, as first postulated by Peeters *et al.* (2009) and demonstrated experimentally by Teng *et al.* (2017)."

*Page 8, line 24: In relation to $HO_x$ production and recycling, the following statement is made about RCIM: "Assuming that photolysis is the dominant fate of the conjugated hydroperoxy-aldehydes (HPALDs) that make up 60% of the stable products, $HO_x$ production can increase . . ." If I understand pp 3352/53 of Wennberg et al. (2018) correctly, the conjugated HPALDs actually only account for 25 % of the products following 1,6 H isomerization; with 15 % unconjugated HPALDs and the remainder other products (e.g. di-HPCARBs). This seems quite different from the stated 60 %. Looking at Fig. 6, it looks like the conjugated species make up 60 % of the total HPALDs, but are 24 % (i.e. 60 % of 40 %) of the full suite of products (with 16 % being the unconjugated species). If this is correct, I presume that the statement on Page 8, line 5 should specify "60% of the HPALDs" rather than "60% of the stable products".*

This sentence was indeed written incorrectly, and we thank the reviewer for their careful attention. The total $HO_x$ production of 3.0 equivalents was meant to represent an upper bound if all HPALDs photolyze. To reflect this, the relevant sentence has been rewritten as follows: "An upper limit of 3.0 equivalents of $HO_x$ production (2.2 OH + 0.75 $HO_2$ + 0.04 $RO_2$) can be achieved in the second oxidative generation if photolysis is also the dominant fate of the HPALDs that make up the remaining 40% of the stable products."

*S4 caption – I think "fun" should be "run".*

This typographic error has been fixed as suggested.

*S19 caption – I think "Jenkin et al. 2015" here is incorrect.*

This copyediting error has been fixed as suggested.

**REVIEWER 2**

*More explanation about the analysis of the fixed radical box modelling is necessary. It is understood that the model is run until complete conversion to $CO_2$ but it is not explained how the concentration for a particular species is calculated; is it the maximum value achieved by a species, the average concentration over a period of time or another metric?*

The fixed-radical box modeling is not used to compute concentrations, but only to quantify product yields from isoprene oxidation by OH. To clarify this, the following sentence has been

added to the "Fixed-radical box modeling" paragraph of Section 2.2: "Product yields are calculated by dividing the total molar production over the entirety of the simulation of each compound of interest by the amount of isoprene oxidized."

*It appears the main aim of the paper is to compare the effect of the new isoprene mechanism with older mechanisms. Therefore, it is felt that Figure 7, which compares global model results of the new isoprene mechanism with a no-isoprene scenario, is much less relevant than Figure S17 in the SI which compares global model results between the new mechanism and the standard GEOS-Chem vn11.02 mechanism. The general effect of isoprene on $NO_x$, $O_3$ and OH is well known in the field. Fig S17 should replace current Figure 7 and the discussion in section 4 should focus more on the differences in global model output between vn11.02 and RCIM rather than RCIM vs. no isoprene. Furthermore, the no-isoprene scenarios plots in Figure 5 should be removed for clarity and more attention paid to the differences between the various mechanisms' outputs.*

Our aim in this manuscript was not primarily to compare RCIM with older mechanisms, but rather to characterize RCIM as a standalone component of box and global models, and to demonstrate the effects of isoprene oxidation (as simulated with RCIM) on regional and global budgets of oxidants, $NO_x$, organic products, and SOA. While the general effects of isoprene on $NO_x$, $O_3$ and OH may be well known in the field, recent changes to our understanding of isoprene oxidative chemistry have forced us to revise some of this knowledge, which merits a new assessment as presented here. We therefore chose to focus primarily on the absolute outcomes following implementation of RCIM into GEOS-Chem, rather than relative changes from the past mechanism. The GEOS-Chem v11-02c mechanism has no intrinsic geophysical value; it represents a snapshot of our chemical understanding of isoprene oxidation at a single point in time, and as such, direct comparisons between the mechanisms will soon be irrelevant when the GEOS-Chem mechanism is updated.

For these reasons, we believe the RCIM-to-no-isoprene maps are better suited for the main manuscript. We still see value in including maps and tables of RCIM-to-v11 differences as a reference, e.g. for previous studies that have used the v11 mechanism for model-measurement comparisons. We also describe the RCIM-to-v11 differences in each subsection of Sections 4 and 5 in the main manuscript. To provide additional information on the differences in global model output between v11 and RCIM as requested by the reviewer, we have added regional product yield statistics from GEOS-Chem v11 simulations to Table S3 in the Supplement, and have added a new table (S6) to the Supplement with inter-mechanism comparisons of global and regional mixing ratios of compounds of interest. We have removed the column from Table S2 that contained similar (but less detailed) information. We hope that these modifications will provide the additional detail and attention to mechanism output differences requested by the reviewer, while maintaining the focus on the RCIM output alone in the main manuscript.

*II would be beneficial to see how the model output using RCIM and vn11.02 compare to observational data. In particular the significant predicted changes to OH, $NO_x$, CO and HCHO*

*over the Amazon and the CO change over much of the southern hemisphere should be compared to observational data if one is to have confidence in the use of RCIM.*

Detailed comparisons to observational data are beyond the scope of this work, and rely heavily on the spatiotemporal distribution of isoprene emissions, a major source of uncertainty on regional scales (see, e.g., Kaiser *et al.*, 2018, and Barkley *et al.,* 2013; we particularly avoid comparisons with formaldehyde observations for this reason, as isoprene emissions are often inferred from formaldehyde retrievals). Because RCIM is built up from chamber experiments and theoretical studies as detailed in Wennberg *et al.* (2018), it provides an independent source of mechanistic detail that should, in theory, not require additional constraint from field observations. However, we agree that some corroboration of the major changes shown in this work with ambient measurements is necessary if future users of RCIM are to have confidence in the mechanism. For this reason, we have included brief comparisons to previous model-measurement studies throughout the text as relevant (e.g., in the original manuscript, p10 L8-11 for ozone in the Southeast US, p11 L28-30 for glyoxal-formaldehyde ratios, p12 L27-30 for MVK/MACR, and p14 L1-3 for isoprene nitrate lifetime).

Observational constraints and prior model-measurement comparison studies have focused largely on the Southeast United States, and our discussion in this manuscript therefore focuses primarily on this region, but as the reviewer rightly points out, many of the largest differences between RCIM and GEOS-Chem v11-02c can be found over the Amazon Basin and in the Southern Hemisphere. While a detailed comparison with observations over the Amazon remains outside the purview of this study, we have added the following passages to relevant sections describing outcomes that exhibit sharp changes over the Amazon:

[to Section 4.1, "Effects on $HO_x$"]: "RCIM increases the simulated annual mean OH concentration over the Amazon by +170% relative to GEOS-Chem v11-02c, and that of $HO_2$ by +30%, both in better agreement with field observations in the region (Barkley *et al.*, 2011)."

[to Section 4.2, "Effects on $NO_x$"]: "For example, global simulations with RCIM result in a 17% increase in annual mean surface $NO_x$ mixing ratios relative to the GEOS-Chem mechanism over the Amazon Basin (see Figure S17 and Table S6), a region where surface $NO_x$ is typically underestimated in GEOS-Chem (Barkley *et al.*, 2011; Liu *et al.*, 2016)."

[to Section 5.1, "Oxygenated VOCs and CO", CO subsection]: "Distributional changes from the GEOS-Chem v11-02c mechanism include 9% higher CO concentrations over the Amazon (due to faster in situ isoprene oxidation from higher OH) and a more diffuse increase of ~2% in CO concentrations throughout the Southern Hemisphere (see Figure S24), where GEOS-Chem tends to underestimate remote surface, column, and upper-tropospheric CO (Zeng *et al.,* 2015; Huang *et al.,* 2016; Fisher *et al.,* 2017)."

*The SOA yield is predicted to be significantly higher than previous models. The contribution to SOA from various species is explained. However, little detail is provided regarding the estimates of SOA production from each species aside from IEPOX. Specifically, the estimates of*

*SOA from HMML, non-IEPOX non-IDHPE species, nitrates, glyoxal and the tetrafunctionalised species are not explained. The decision to treat the tetrafunctionalised species as LVOCs within the GEOS-Chem framework also warrants further discussion as the species span a wide range of volatilities.*

Substantial updates to the complex SOA formation scheme in GEOS-Chem are beyond the scope of this work, and will be the subject of a future study with more detailed treatment of reactive uptake parameterizations (e.g. Jo *et al.*, 2019), incorporation of additional reactive and depositional particle sinks (e.g. Hodzic *et al.*, 2016), and comparisons to field measurements (e.g. Pai *et al.*, 2019). For this reason, we chose to focus primarily on the production of gas-phase SOA precursors in Section 5.3 of the present manuscript, and we direct the reader to Marais *et al.* (2016) for more detail on the SOA uptake parameterizations of each individual compound in Sections 2.1 and 5.3. We describe the gas-phase production of HMML, non-IEPOX non-IDHPE species, and the tetrafunctional species in detail in Section 5.3, and in the same section briefly describe the SOA-relevant production of glyoxal (for which more detail can be found in Section 5.1) and organonitrates (the subject of Section 5.2). We agree that the uniform SOA uptake treatment of the tetrafunctional species as identical to the "LVOC" species already contained in GEOS-Chem is overly simplistic. The uptake parameterization will need to be amended as further constraints become available, and for immediate implementation in GEOS-Chem, it should be reduced to bring SOA formation in line with previous model-measurement comparisons. To highlight these points, we have a number of clauses and sentences to the C5 tetrafunctional compound subsection of Section 5.3:

... "MCM and GEOS-Chem v11-02c predict similar yields of $C_5$ tetrafunctional species, but the relative contributions of individual species vary substantially between mechanisms (See Figures S22-23). GEOS-Chem v11-02c only considered SOA formation from two such species (dihydroxy-dinitrates and "LVOC" produced in the reaction of ISOPOOH with OH), resulting in 4 Tg a$^{-1}$ iSOA from $C_5$ tetrafunctional compounds. Because the rates of gas-phase oxidation, deposition, and aerosol uptake for these compounds are all poorly constrained, their contribution to iSOA remains highly uncertain, and future studies will need to evaluate the volatilities, solubilities, and particle-phase reactivities of the individual tetrafunctional species."
... "This total carries high uncertainty, due both to the SOA uptake parameterization and the lack of constraints on other loss pathways of the $C_5$ tetrafunctional compounds, but is similar to a recent estimate by Stadtler *et al.* (2018) and highlights the importance of further investigations of this iSOA formation pathway. Until such studies are performed, we recommend reducing the LVOC uptake coefficient applied to the tetrafunctional species by a factor of ten in GEOS-Chem implementations, to bring iSOA production from this pathway in line with previous model-measurement comparisons (Marais *et al.*, 2016; Pai *et al.*, 2019)."

**MINOR ADDITIONAL CHANGES**

Several small changes have been made throughout the manuscript for clarity, detailed below with page and line numbers corresponding to the discussion manuscript:

[P5L6] "NO$_2$/NO molar ratio of 5,"

[P5L17] "NO emissions are constant for a given simulation and are varied between simulations to diagnose the sensitivity of the isoprene oxidation cascade to NO$_x$; results are presented as a function of the daytime NO$_x$ concentration."

[P6L18] "Nighttime oxidation by NO3 is particularly lower than previously reported in the literature (5-7% globally, Table 1), which largely reflects the amount of isoprene remaining at sunset. More efficient
recycling of OH in RCIM  results in less isoprene at sunset."

[P8L18] "an increase in mean  daytime temperature of 10 °C causes up to a doubling in  OH concentrations"

[P10L20] "The  remaining 13% forms isoprene SOA, which  is assumed in GEOS-Chem to have no further chemical reactivity (Marais et al., 2016)"

[P11L12] "Fixed-radical box model simulations with RCIM"

[P11L19] "the overall global molar yield of formaldehyde from isoprene, which we estimate to be 111% (22% per carbon, a 4% increase from GEOS-Chem v11-02c). The yield is lower than in the box model simulations of Figure 10 because of deposition and aerosol uptake of isoprene oxidation intermediates."

[P16L3] "RCIM results in similar production of HMML  as in MCM"

[P16L22] "The organonitrate iSOA formation simulated in GEOS-Chem is therefore likely an upper limit ."

[P16L27] "production of iSOA from glyoxal is 10% of that from IEPOX. Locally, however, glyoxal can still be an important contributor"

[P17L23] "ozone, CO, and formaldehyde concentrations between the two mechanisms"

[P18L7] "RCIM estimates a higher fraction of isoprene reacting with OH globally (88%) than past mechanisms.  The resulting hydroxy-peroxy radicals (ISOPOO) react  with HO$_2$ (41%), NO (28%), and RO$_2$ (9%), or undergo H-shifts to regenerate HO$_x$ (22%)."

[P18L21] "76% of which proceeds via CO  including 44% via formaldehyde"

[p18L30] "This 13% SOA yield per carbon (25% yield by mass)"

The caption of Figure 3 was revised for clarity: "daytime mean $NO_x$ concentration and temperature"

The caption of Figure 4 was revised for clarity: "Percent of isoprene  reacting with $O_3$ and $NO_3$, and percent of the products from the reaction of isoprene with OH (ISOPOO hydroxy-peroxy radicals)  reacting via each pathway.

The caption of Figure 6 was revised for clarity: " non-radical (closed-shell) products are shown in blue"

The caption of Figure 8 was revised for clarity: "(c)  isoprene-derived SOA production."

The caption of Figure 10 was revised for clarity: "Percent yields of organic products from isoprene + OH oxidation as a function of NO and $HO_2$ … Contours are evenly spaced on a linear scale between the  minimum values (in white) and maximum values (in black) located on each plot."

The citation for Wolfe et al. (2016) referred to the incorrect 2016 paper by Glenn Wolfe; the relevant entry in the reference list has been updated.

**REFERENCES**

[revised manuscript text omitted]

[a]Results from $2° \times 2.5°$ horizontal resolution global simulations. Positive percentages indicate higher mixing ratios in GEOS-Chem v11-02c than RCIM. [b]Average mixing ratios from 0-1 km altitude. [c]Causing a 1.16% decrease in the tropospheric methane lifetime. [d]
[revised manuscript text omitted]